# Mechanistic patterns and clinical implications of oncogenic tyrosine kinase fusions in human cancers

Taek-Chin Cheong [1,7] ✉, Ahram Jang [1,2,7], Qi Wang [1], Giulia C. Leonardi[1,3], Biagio Ricciuti[4], Joao V. Alessi[4], Alessandro Di Federico[4], Mark M. Awad[4], Maria K. Lehtinen [1], Marian H. Harris [1] & Roberto Chiarle [1,5,6] ✉

Tyrosine kinase (TK) fusions are frequently found in cancers, either as initiating events or as a mechanism of resistance to targeted therapy. Partner genes and exons in most TK fusions are followed typical recurrent patterns, but the underlying mechanisms and clinical implications of these patterns are poorly understood. By developing Functionally Active Chromosomal Translocation Sequencing (FACTS), we discover that typical TK fusions involving ALK, ROS1, RET and NTRK1 are selected from pools of chromosomal rearrangements by two major determinants: active transcription of the fusion partner genes and protein stability. In contrast, atypical TK fusions that are rarely seen in patients showed reduced protein stability, decreased downstream oncogenic signaling, and were less responsive to inhibition. Consistently, patients with atypical TK fusions were associated with a reduced response to TKI therapies. Our findings highlight the principles of oncogenic TK fusion formation and selection in cancers, with clinical implications for guiding targeted therapy.

Tyrosine kinase (TK) gene fusions are common genetic alterations across the cancer types, including both hematologic and solid cancers. They are one of the earliest genomic events that initiate oncogenesis, as demonstrated in functional models[1], as well as in cancer genome studies[2-4]. Furthermore, acquisition of TK fusions, such as ALK or RET fusions, have been also reported during targeted therapy in non-small cell lung cancer (NSCLC) and other tumors, as a mechanism of resistance[5-9]. Identification of the functional TK fusions is crucial in the clinic because small-molecular TK inhibitors are highly effective for patients with cancers harboring these TK fusions, often regardless of the tissue of origin[10-12].

TK genes typically fuse with a partner gene that provides an active promoter for the fusion gene's expression and dimerization or oligomerization domains for TK activation through the fused TK domain[13]. Mechanistically, TK fusions are often formed by genomic rearrangements between two DNA double-strand breaks (DSBs) in introns, leading to the transcription of in-frame chimeric gene products. However, the mechanisms by which these recurrent breakpoints are selected among the large pool of potential fusion combinations remain unclear[14]. For example, in patients with NSCLC, *EML4* is the most frequent partner gene of *ALK* fusions and *CD74* for *ROS1* fusions. However, *EML4-ROS1* or *CD74-ALK* fusions have not been reported,

[1]Department of Pathology, Boston Children's Hospital and Harvard Medical School, Boston, MA 02115, USA. [2]Division of Endocrinology, Diabetes, and Metabolism, Department of Medicine, Beth Israel Deaconess Medical Center, Boston, MA 02115, USA. [3]Department of Biomedical and Biotechnological Sciences, University of Catania, 95123 Catania, Italy. [4]Lowe Center for Thoracic Oncology, Dana-Farber Cancer Institute, Boston, MA 02115, USA. [5]Department of Molecular Biotechnology and Health Sciences, University of Torino, Torino 10126, Italy. [6]Division of Hematopathology, IEO European Institute of Oncology IRCCS, 20141 Milan, Italy. [7]These authors contributed equally: Taek-Chin Cheong, Ahram Jang. ✉e-mail: taekchin.cheong@childrens.harvard.edu; roberto.chiarle@childrens.harvard.edu

although these fusions can theoretically be functional. Furthermore, multiple introns in *ALK* can potentially create in-frame *ALK* fusions fully preserving the kinase domain, but most *ALK* fusions involve breaks in intron 19, regardless of its partner genes[15,16]. The molecular basis of selecting partner genes, introns, and the clinical implications of different fusion types between the typical and atypical fusions remain unclear.

In this study, we identify mutually exclusive fusion partner selection and specific exon usage in TK fusions based on the Catalogue of Somatic Mutations in Cancer (COSMIC) datasets. We develop an experimental framework integrating high-throughput genome-wide gene fusion sequencing followed by clonal selection under the pharmaceutical selective pressure, which we call Functionally Active Chromosomal Translocation Sequencing (FACTS). Through this approach, we identify oncogenic TK fusions that spontaneously occur in the NSCLC cells and confer selective advantages. Furthermore, we determine the critical role of gene transcription and protein stability to explain the recurrent selection of the typical TK fusions. Finally, we highlight their clinical implications that impact the outcome of patients during TKI treatment.

## Results

### Characterization of kinase fusions across cancer types

We analyzed 8,805 3' kinase gene fusions curated from the COSMIC, focusing on the seven most common kinase fusions involving *ALK*, *RET*, *ROS1*, *NTRK1*, *NTRK3*, *ABL1*, and *BRAF* genes found in various types of cancers (Supplementary Data 1). In this study, we used 7,751 kinase fusions where the mRNA junction positions were validated (Supplementary Data 1).

We first analyzed the prevalence of the seven 3' kinase fusions across 16 tissue types. Except for ABL1 fusions, all kinase fusions were identified in multiple tissue types at variable frequencies between the tissue types (Fig. 1a and Supplementary Data 2). We observed several common patterns of kinase fusions, in terms of partner genes and intron usage. ALK fusions were predominantly observed in lung cancer and lymphoma, while RET, ABL1, and BRAF fusions have been identified most frequently in thyroid cancer, leukemias, and pediatric low-grade gliomas, respectively. *EML4* was the most frequent partner of *ALK* fusions in lung cancers, while *NPM1* was the case in lymphomas (Fig. 1b and Supplementary Data 1). The other TK fusions, including *RET*, *ROS1*, *BRAF*, and *ABL1* fusions, showed several frequent partner genes (Fig. 1b and Supplementary Data 3). Partner genes were largely specific to kinase genes, with several exceptions (Fig. 1c). For example, seven partner genes (9.6%), including *KIF5B*, *TPM3*, *ERC1*, and *ETV6*, were shared across the 73 kinase fusions (Fig. 1c, d and Supplementary Data 1). In contrast, most frequent partner genes, i.e., *EML4*, *CCDC6*, *CD74*, *BCR* and, *KIAA1549* were exclusively associated with *ALK*, *RET*, *ROS1*, *ABL1* and *BRAF* fusions, respectively (Fig. 1c, d and Supplementary Data 3). In summary, this analysis shows tissue type- and kinase gene-specific partnering in fusion oncogene formation.

Next, we analyzed the locations of fusion events between the kinase and partner genes. Notably, most fusion events in *ALK*, *RET*, and *ABL1* occurred at the 5' end of exon 20 (e20; 99.6%, 837/839), e12 (99.5%, 1,047/1,052), and e2 (99.1%, 5,040/5,087), respectively, regardless of their fusion partners. In contrast, *ROS1* fusions occurred at variable locations, including e32 (20.5%), e34 (53.8%), e35 (14.1%), and e36 (11.5%) (Fig. 1e and Supplementary Data 3). A similar pattern was observed in the partner genes. Several partner genes showed exclusive preference in exon usage. For example, all *NPM1-ALK* and *CD74-ROS1* fusions used a specific exon in the partner genes (e5 of *NPM1* and e6 of *CD74*). In contrast, *EML4* and *KIF5B* used several exons, including e6, e13, and e20 for *EML4* and e15 and e16 for *KIF5B* (Fig. 1f and Supplementary Data 3). This indicates potential selective benefit of specific exon usage in creating kinase fusions.

## Functionally Active Chromosomal Translocation Sequencing (FACTS) identifies oncogenic ALK fusions genome-wide

Genome-wide techniques identifying chromosomal translocations, such as HTGTS and TC-seq, have been widely used to study the mechanisms of translocation in normal and tumor cells by cloning chromosomal junctions generated within a few days of inducing a programmed DNA DSB at a specific site[17–19]. However, not all these translocations result in functional fusion genes, and their impact on oncogenesis cannot be determined by these technologies alone. To overcome this limitation, we developed FACTS to specifically map functional oncogenic fusions genome-widely in the setting of pharmaceutical selective pressure (Supplementary Fig. 1). Inspired by the recent reports indicating fusion oncogene formation as a mechanism of resistance to EGFR inhibitors[7,20,21], we applied FACTS into the in vitro model of EGFR inhibitor resistance using PC-9 cells, harboring *EGFR*-activating mutation (*EGFR* E746-A750del)[22].

PC-9 cells have been extensively used to characterize various mechanisms of resistance to EGFR inhibitors. Under the EGFR inhibitor treatment, PC-9 cells typically undergo cell-cycle arrest, and a small number of cells undergo persistence. Emergence of fully resistant clones require additional genetic alterations, including secondary mutations in EGFR that prevent TKI binding[23] or another oncogenic driver events that bypasses EGFR inhibition, such as MET amplifications or the acquisition of fusion oncogenes[24]. Therefore, we reasoned that PC-9 cells under selective pressure from a selective EGFR inhibitor osimertinib[7,25] would be an ideal setting to test the functional outcome of various kinase fusions induced by genome-wide translocations.

ALK fusions are the most frequent TK fusions found in 3-7% of patients with NSCLC[15] and drive resistance to targeted therapy in patients with EGFR-mutant or KRAS[G12C]-mutant cancer[7–9]. Consistently, PC-9 cells expressing two sgRNAs targeting the relevant introns in *EML4* (intron 6 or intron 13) and *ALK* (intron 19) to force the formation of typical EML4-ALK fusions generated resistant clones expressing the EML4-ALK fusions at a frequency of ~1 ×10⁴ clones/million cells (1.16%-1.34%) upon osimertinib selection (Supplementary Fig. 2a-d), which was consistent with the frequency of translocations induced by two DNA DSBs in previous studies[26,27]. Furthermore, *EML4-ALK* fusion expressed from the endogenous *EML4* promoter rapidly induced osimertinib resistance in PC-9 cells by producing active and phosphorylated EML4-ALK fusion proteins (Supplementary Fig. 2e). Next, we applied FACTS to study ALK fusion formation from a single programmed DSB with any partners in the genome under the osimertinib treatment. We induced a programmed DSB in intron 19 of *ALK*, the hotspot of genomic rearrangements causing EML4-ALK fusions in NSCLC[15] (Fig. 2a). Multiple osimertinib-resistant clones developed during selection, with an estimated frequency of ~ 0.05 or 1 clone/million cells by targeting only *EML4* intron 6 or *ALK* intron 19, respectively (Fig. 2b). Analysis of single clones showed that each expressed ALK proteins with a wide range of sizes from ~60 kDa to 400 kDa, likely indicating formation of fusion oncoprotein in different sizes (Supplementary Fig. 2f). In contrast, when we introduced a DSB in intron 6 of *EML4* gene, no ALK expression was observed in osimertinib-resistant clones (Supplementary Fig. 2g), indicating that the DSB at *ALK* is critical in fusion formation. The ALK fusions formed after the DSBs at *ALK* intron 19 showed strong phosphorylation of the kinase domain, resulting in sustained activation of the MAPK pathway despite the presence of osimertinib, but showed impaired phosphorylation of the kinase domain in the presence of ALK-specific inhibitor lorlatinib, resulting in decreased activation of the MAPK pathway (Fig. 2c), likely explaining the mechanism of resistance. The combination of osimertinib with lorlatinib completely blocked ALK and ERK1/2 phosphorylation (Fig. 2c). Consistently, the growth of osimertinib-resistant clones was inhibited by the combination of osimertinib with lorlatinib, but not by osimertinib or lorlatinib alone (Fig. 2d). These data showed

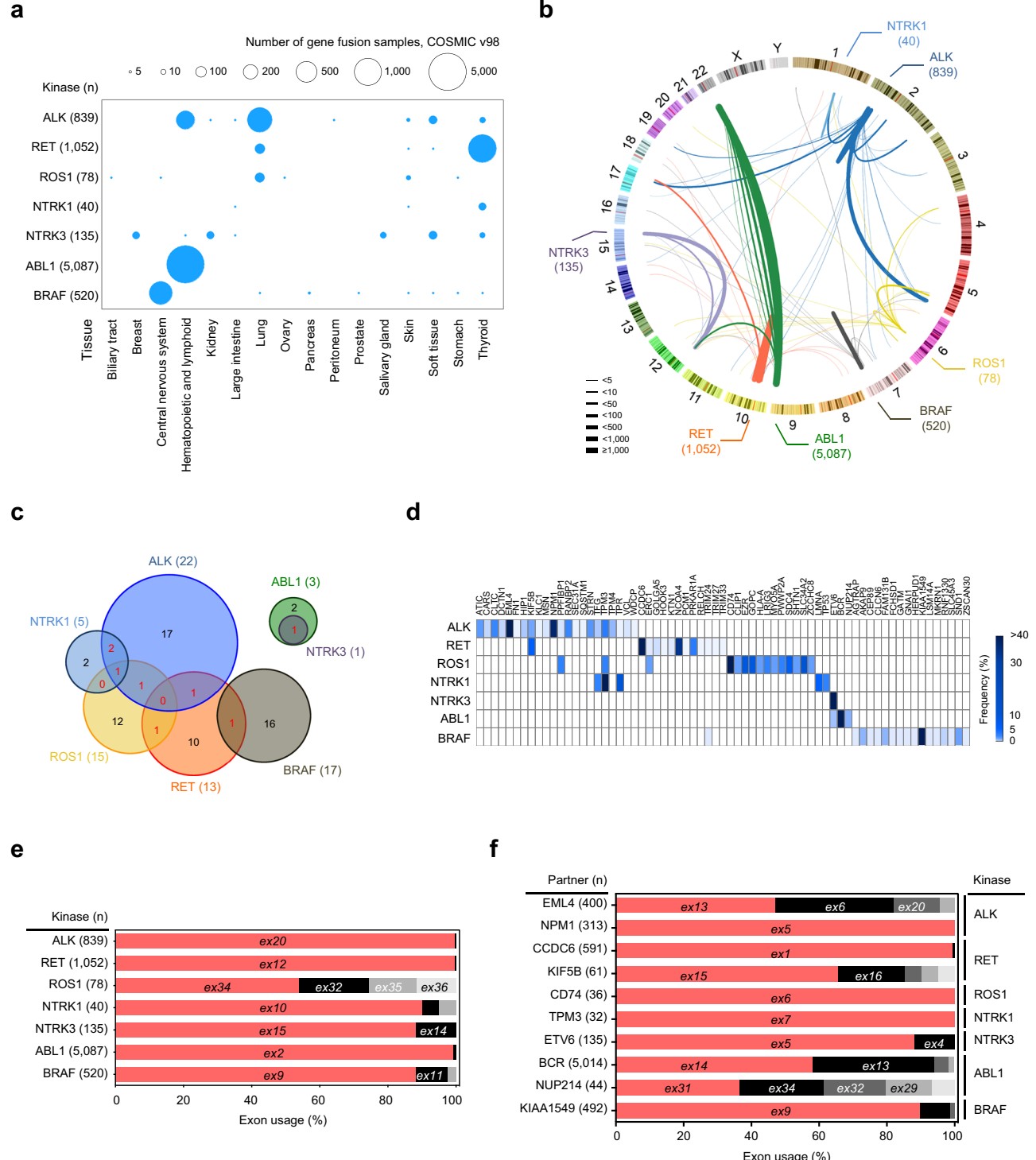

**Fig. 1 | Landscape and characterization of recurrent 3′ kinase fusions across multiple cancers curated from COSMIC v98. a** The dot plot indicates the distribution of kinase fusion in each tissue type. The size of the dot corresponds to the number of kinase fusions in each tissue type. **b** Circos plot showing the genome-wide distribution of 3′ kinase fusions. The thickness of arcs represents the number of fusions of each kinase fusions, and each color corresponds to fusions with each kinase gene. **c** Venn diagrams showing the overlap of fusion partners shared between *ALK* (n = 22 partner genes), *NTRK1* (n = 5), *ROS1* (n = 15), *RET* (n = 13), *BRAF* (n = 17), *ABL1* (n = 3), and *NTRK3* (n = 1) fusions. Each color corresponds to each kinase gene. **d** Frequency of kinase fusion partners. **e, f** Exon usages of kinase genes (**e**) and partner genes (**f**) in each 3′ kinase fusions. Source data are provided as a Source Data file.

that a DSB in *ALK* led to the formation of in-frame fusions, of which expression conferred resistance to osimertinib.

Next, we identified unknown 5′ partner genes of these *ALK* fusions by using a 3′ end-directed fusion assay[28]. Several in-frame *ALK* fusions were identified (Fig. 2e and Supplementary Data 4), and a subset of

them was further validated in single clones using RT-PCR (Supplementary Fig. 2h). Three of these fusion partners, *EML4, STRN*, and *ATIC*, are on chromosome 2, where *ALK* is located, whereas others were spread in the genome (Fig. 2e and Supplementary Fig. 2i). Remarkably, several of these spontaneous ALK fusions were identical to those

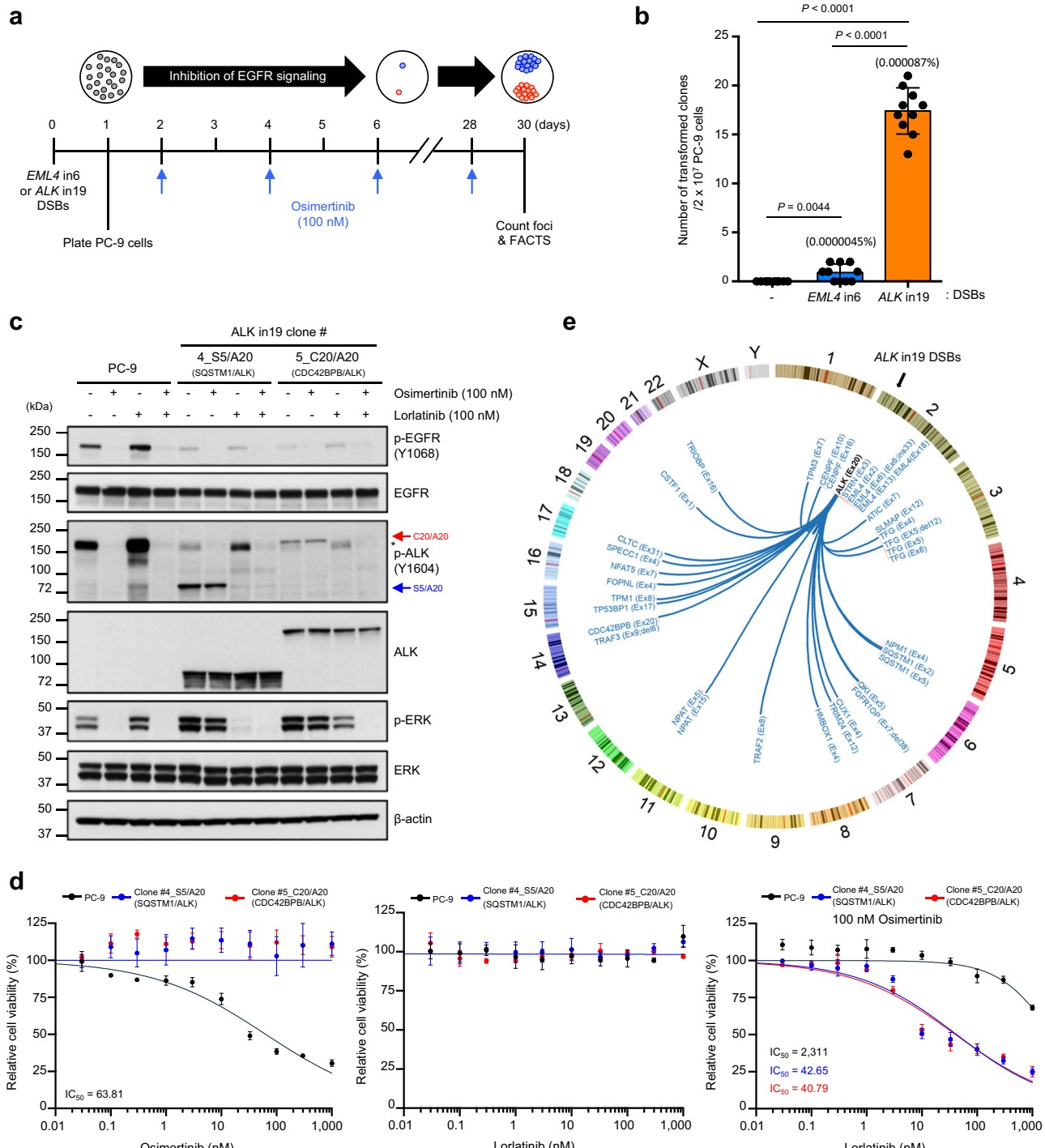

**Fig. 2 | Spontaneously formed ALK fusions induce resistance to osimertinib in PC-9 cells. a** Experimental timeline to select osimertinib-resistant clones in PC-9 cells. **b** Quantification of osimertinib-resistant clones in PC-9 cells treated as in (**a**). Data show means of ten biological replicates, with error bars representing ±s.e.m; significance was determined by an unpaired, two-tailed Student's $t$-test. **c** Representative western blots of the signaling changes in osimertinib-resistant clones treated with EGFR inhibitor (osimertinib) and/or ALK inhibitor (lorlatinib). Red and blue arrows represent phosphorylated CDC42BPB-ALK and SQSTM1-ALK

fusion proteins, respectively. Asterisk represents a non-specific band. Similar results were observed in $n = 2$ independent experiments. **d** Sensitivity to osimertinib (**left**), lorlatinib (**middle**), and combination of osimertinib plus lorlatinib (**right**) in osimertinib-resistant clones. Data show means of three biological replicates, with error bars representing ±s.e.m. **e** Circos plot showing the genome-wide distribution of ALK fusion partners identified in osimertinib-resistant PC-9 cells. Arcs represent functional rearrangements joining *ALK* (exon 20) to the indicated fusion partner. Source data are provided as a Source Data file.

described in NSCLC[29] or in other tumor types[4] (Supplementary Data 4). For example, *EML4-ALK* fusions joined e2, e6, e13, or e18 of the *EML4* gene to e20 of the *ALK* gene (Fig. 2e and Supplementary Data 4), exactly as seen in patients with NSCLC and other tumors[30,31]. In addition, some fusion partners showed exclusive exon usage, such as e3 in

*STRN* and e31 in *CLTC*, while others showed variable usage (Fig. 2e and Supplementary Data 4). These exon usages were identical to the corresponding ALK fusions found in NSCLC and thyroid cancer (Supplementary Data 4). We also identified a list of ALK fusions that have not been described in human tumors (Supplementary Data 4), which may

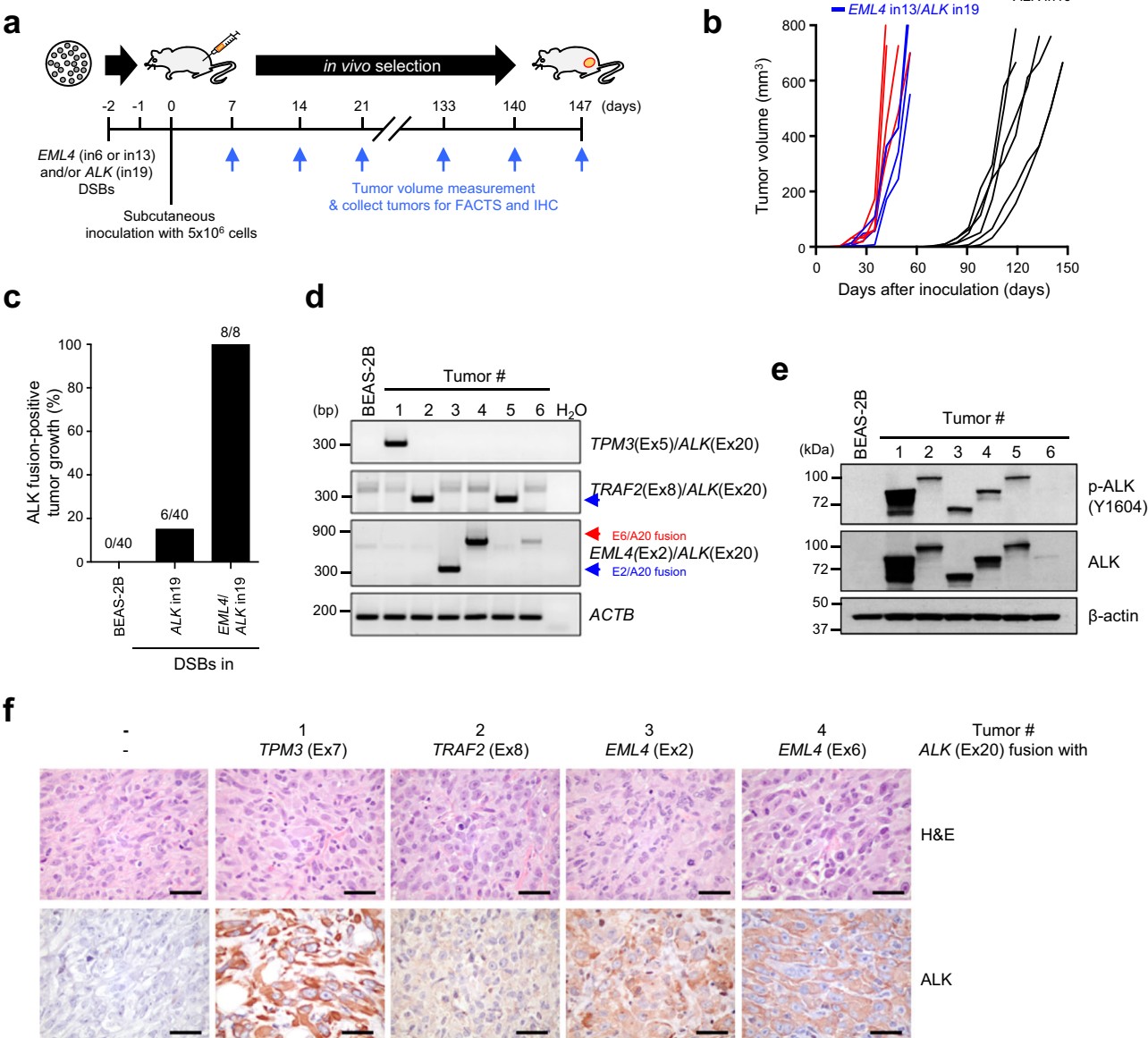

**Fig. 3 | Spontaneously formed ALK fusions drive tumor formation in vivo.**
**a** Experimental timeline of the in vivo experiment in NSG mice grafted with BEAS-2B cells. **b** Tumor growth in NSG mice grafted with BEAS-2B cells. DNA DSBs in intron 6 of *EML4* and intron 19 of *ALK* (red, $n = 4$), in intron 13 of *EML4* and intron 19 of *ALK* (blue, $n = 4$), or in intron 19 of *ALK* (black, $n = 40$). **c** Percentage of ALK-fusion positivity in tumors grown in NSG mice grafted with BEAS-2B cells. **d** Validation of the indicated *ALK* fusion transcripts from tumors developed in NSG mice. Similar results were obtained from $n = 2$ independent experiments. **e**, **f** Representative western blots (**e**) and immunohistochemistry (**f**) from tumors grown in mice grafted with BEAS-2B cells introduced DNA DSBs only in intron 19 of *ALK*. Scale bar = 50 μm. Similar results were obtained from $n = 2$ independent experiments. Source data are provided as a Source Data file.

indicate rare functional fusion events yet to be discovered. Several of them (e.g., *QKI*, *TRAF2*, *TRAF3*, and *TP53BP1*) were the genes previously reported in fusions with other kinases[32–34] and contain dimerization or oligomerization domains, further supporting their functionality (Supplementary Fig. 2j and Supplementary Data 4). Taken together, FACTS demonstrated functional fusion oncogene formation through genome-wide translocations after a single DSB at *ALK* intron 19 and reproduced *ALK* fusion landscape in human cancers.

To test whether oncogenic ALK fusions can also be generated in non-cancerous cells, we applied FACTS to the bronchial epithelial BEAS-2B cells. These cells can grow in vivo once they are transformed by oncogenic drivers[35]. As a positive control, we injected mice with BEAS-2B cells where two DSBs were induced in *EML4* intron 6 or 13 and *ALK* intron 19. As expected, all mice in this group developed tumors (Fig. 3a–c). When we injected mice with BEAS-2B where a

single DSB was introduced in *ALK* intron 19, we observed tumor formation at a lower rate and with a slower growth kinetics (Fig. 3b, c). FACTS and RT-PCR validation revealed that tumors expressed various ALK fusions identical to those in PC-9 cells and patient samples (Fig. 3d and Supplementary Data 4). Protein expression of these fusions was most likely determined by the fusion partner, with some fusions being expressed at higher levels than others (Fig. 3e, f). Thus, by applying FACTS to immortalized normal-like bronchial epithelial cells, we demonstrated that a single DSB in *ALK* intron 19 produced functional ALK fusion oncogenes that led to a malignant transformation in vivo.

### FACTS identifies oncogenic RET, ROS1, and NTRK1 fusions
We next applied FACTS to other kinase fusions. *RET, ROS1*, and *NTRK* family gene fusions are found in ~4% of patients with NSCLC[36].

We designed FACTS by introducing one DSB in their intron that is most frequently involved in chromosomal translocations, (i.e., intron 11 for *RET*, intron 33 for *ROS1*, and intron 11 for *NTRK1*; Supplementary Fig. 3a–d). Resistant clones developed after 4 weeks of osimertinib selection at a frequency comparable to what was observed from clones with ALK fusion (Fig. 2b and Supplementary Fig. 3e). FACTS identified several in-frame chimeric proteins with RET, ROS1, and NTRK1 and their joined partners across the genome (Supplementary Fig. 3f–n and Supplementary Data 4). We validated some of these acquired fusions by RT-PCR and Sanger sequencing and confirmed that they were identical to the *RET* fusions described in patients with NSCLC (Supplementary Fig. 3o, p). Other fusions were not yet described in patients (Supplementary Data 4). We confirmed that acquired RET fusions conferred resistance to osimertinib, as demonstrated by the reversal of resistance phenotype by selpercatinib (Supplementary Fig. 3q). The fusion partner genes identified here also contained dimerization or oligomerization domains (Supplementary Fig. 3i–k and Supplementary Data 4). Intriguingly, FACTS identified exon fusion variants involving different *ROS1* exons (e34, e35, and e36) or *NTRK1* exons (e12 and e13) but only e12 of *RET* (Supplementary Fig. 3f–h and Supplementary Data 4), which is consistent with what was reported from patients[29,37–39].

## Gene transcription, rather than chromatin accessibility, dictates the selection of partner genes in TK fusions

Next, we investigated how ALK fusions are selected among many potential rearrangements, and which mechanistic factors dictate the choice of ALK fusion partners in the genome. Among all reported partner genes of ALK fusion from the patients with NSCLC, those identified by FACTS in our PC-9 model showed a significantly higher level of transcription (Supplementary Fig. 4a, b). In contrast, we found no difference in terms of chromatin accessibility measured by ATAC-seq or histone activation marks between the FACTS-identified and -unidentified genes (Supplementary Fig. 4c, d). Recurrent translocation partners consistently showed active chromatin marks (Supplementary Fig. 4e-j). These findings were consistent in partner genes of RET, ROS1, and NTRK1 fusions[37–39] (Supplementary Fig. 4k–p). Taken together, the partner genes selected in FACTS were associated with higher level of transcription, compared to the other partner genes not identified by FACTS but reported in patients.

Next, we further explored whether gene transcription was sufficient to induce the formation of oncogenic fusions. PC-9 cells express very low to undetectable levels of HLA-DR molecules and the invariant chain CD74 that is essential for the assembly and subcellular trafficking of the MHC class II complex[40] (Supplementary Fig. 5a-d). We hypothesized that this undetectable expression could explain why CD74 or HLA-DR fusions with kinases[41] were not identified by FACTS in PC-9 cells. Because expression of both CD74 and HLA-DR can be induced by the Class II transactivator (CIITA)[42] (Supplementary Fig. 5e), we asked whether induction of *HLA-DR* or *CD74* expression by CIITA was sufficient to generate fusions of HLA-DR or CD74 with ROS1. FACTS was applied to PC-9 cells expressing CIITA that showed significantly increased HLA-DR and CD74 mRNA and protein levels (Supplementary Fig. 5f–m). By introducing DSBs in intron 33 of *ROS1* (Fig. 4a), we identified genome-wide oncogenic fusions including HLA-DRB1-ROS1 fusions in which the breakpoint in the *HLA-DRB1* gene was identical to that observed in patients with HLA-DRB1-MET fusion[41] (Fig. 4b, c and Supplementary Data 5). We estimated the frequency of HLA-DRB1-ROS1 fusions at 6.7% using single clone analysis (Fig. 4d). In contrast to HLA-DRB1-ROS1 fusions, CD74-ROS1 fusions were not detected, which suggests that induction of transcription for the *CD74* gene was not sufficient to trigger CD74-ROS1 translocations. However, when we simultaneously introduced DSBs in both *CD74* and *ROS1* genes in either control PC-9 or CIITA-expressing PC-9 cells, resistant clones rapidly emerged only in CIITA-expressing PC-9 cells (Fig. 4e, f). While DNA

junctions were detected in both cells, CD74-ROS1 fusion transcripts were detected only in CIITA-expressing PC-9 cells (Fig. 4g-i). These results suggest that increased gene expression of the partner gene is sufficient to induce the formation of TK fusions in loci such as HLA-DRB1, which is located on the same chromosome with ROS1, and point out that the detection of DNA junctions is insufficient to determine oncogenicity of resulting TK fusions without evidence of efficient transcription of the TK fusion.

## Oncogenic TK fusions originate after selection of pools of rearrangements spontaneously occurring in fusion partner and TK genes

In some tumors such as lymphoma, recurrent translocations are the result of the activity of the activation-induced cytidine deaminase (AID) enzyme that targets specific regions of the genome[43]. Therefore, we asked whether the selection of specific partners or exons by TK fusions is mechanistically determined by the formation of DSBs in specific positions of genes or rather by the selection of random genomic DSBs. To this end, we generated libraries of DNA junctions using HTGTS, allowing for an unbiased detection of genome-wide chromosomal rearrangements[17], in PC-9 cells introduced a programmed DSB in intron 19 of *ALK* in both before and after selection (Supplementary Fig. 6a, b). HTGTS yielded 111,811 genomic translocation breakpoints before selection, which were distributed throughout the genome with enhanced clustering in the 2 Mbp regions surrounding the *ALK* DSB (Fig. 5a–c and Supplementary Fig. 6c). We identified 154 hotspots with significantly enriched breakpoint clustering (Fig. 5d and Supplementary Data 6). Only 2.6% (4/154; *EML4*, *SQSTM1*, *TRAF2*, and *CLTC*) of these hotspots occurred in genes that are known partners of ALK fusions (Fig. 5d and Supplementary Data 6). In sharp contrast, HTGTS performed with resistant clones after osimertinib selection yielded 5005 DNA breakpoints with 81% hotspots (13/16) occurring in genes leading to the transcribed ALK fusions identified by FACTS (Fig. 2e and Fig. 5a and Supplementary Data 4 and 6). Several strong genomic translocation hotspots observed before selections completely disappeared after selection (Fig. 5e–g and Supplementary Fig. 6d), most likely because the resulting rearrangements did not generate a functional *ALK* fusions. *EML4* was the gene most frequently translocated with *ALK* after selection (Fig. 5h). Breakpoints before selection did not show a preferential strand bias, which is consistent with previous works of genome-wide cloning of unselected translocations[17,44]. In contrast, the breakpoints after selection showed a strong bias for an orientation of the gene leading to a functional fusion with *ALK* (Fig. 5i), with DNA breakpoints markedly enriched for junctions occurring in gene introns (Fig. 5j). Within individual partner genes, we observed a selective enrichment of breakpoints occurring in introns leading to in-frame functional fusions with ALK (Fig. 5k and Supplementary Fig. 6e-h). Overall, these data indicate that the formation of ALK fusion is the result of a functional selection of transcribed translocations based on the location and orientation, not just a reflection of DNA break frequency.

Next, we focused on TK genes and generated HTGTS libraries in BEAS-2B and PC-9 cells by inducing a DSB in *EML4* as bait to capture breaks spontaneously occurring in TK genes (Supplementary Fig. 7a). We looked at the distribution of breakpoints in *ALK*, *RET*, *ROS1*, *NTRK1*, as well as other kinase genes known to generate oncogenic fusions in lung cancer, such as *EGFR*, *ERBB4*, *MET*, *FGFR3*, and *EPHA2*[45]. Breakpoints identified in these kinases were spread throughout the gene body including introns and exons without clear clusters (Supplementary Fig. 7b–j). In both BEAS-2B and PC-9 cells, more breakpoints were observed in *ALK* than in other kinases (Supplementary Fig. 7k, l and Supplementary Data 7), most likely because *EML4* and *ALK* are proximally located on the same chromosome 2[46,47]. More breakpoints in *EGFR* were detected in PC-9 cells than in BEAS-2B cells (Supplementary

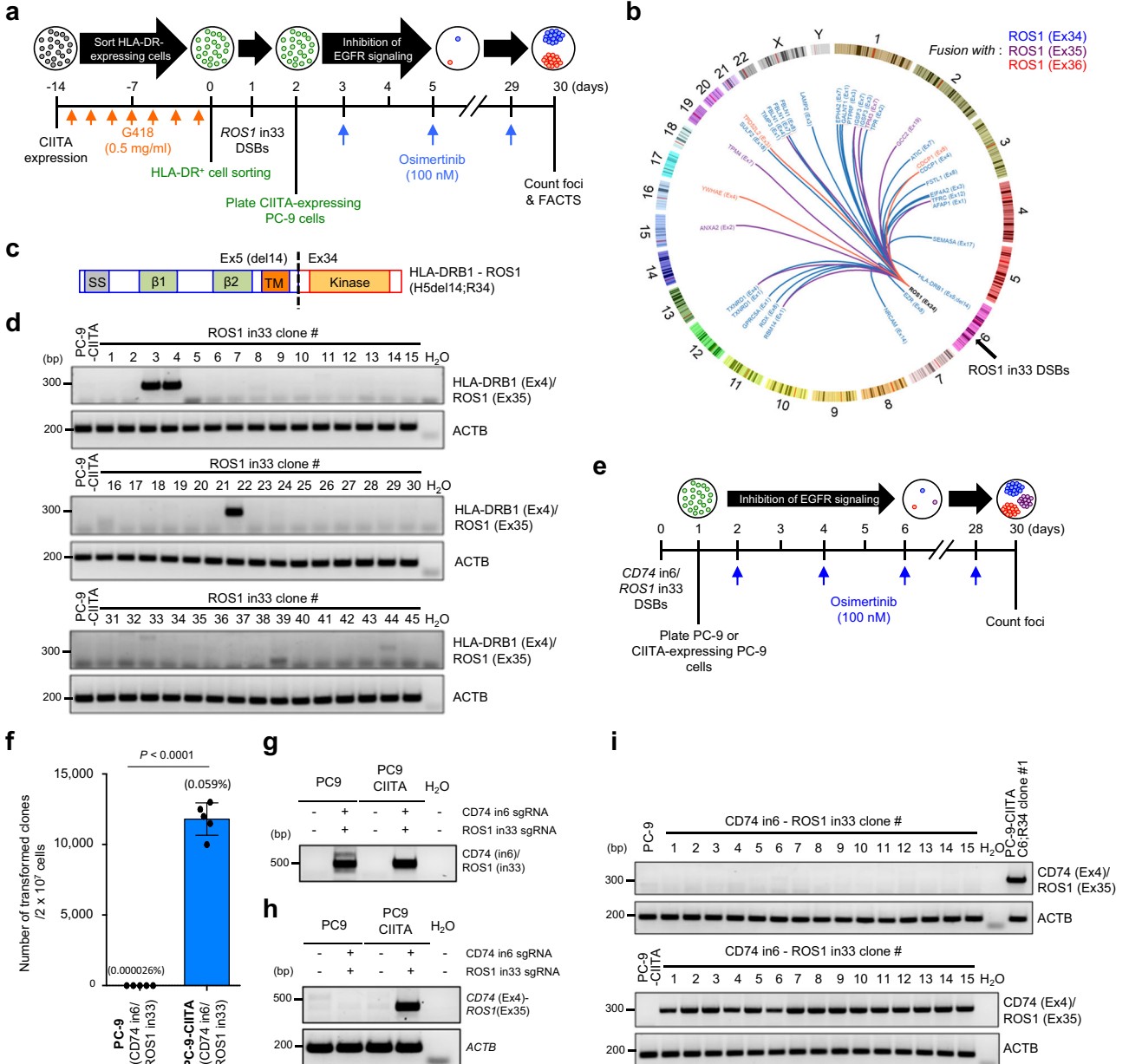

**Fig. 4 | Reactivation of transcription by CIITA is sufficient to generate functional HLA-DRB1-ROS1 and CD74-ROS1 fusions. a** Experimental design to select osimertinib-resistant clones in CIITA-expressing PC-9 cells. **b** Circos plot showing the genome-wide distribution of *ROS1* fusion partners identified in osimertinib-resistant cells obtained as described in (**a**). Arcs represent functional rearrangements joining *ROS1* (exon 34, 35 or 36) to the indicated fusion partner. **c**, Schematic structural composition of HLA-DRB1-ROS1 H5del14;R34 fusion proteins. Protein domains are indicated by color and include: SS, signal sequence; TM, transmembrane domain; Kinase, ROS1 tyrosine kinase domain. **d** Detection of HLA-DRB1-ROS1 transcripts in CIITA-expressing PC-9 cells. **e** Experimental design to select

osimertinib resistant clones driven by CD74-ROS1 fusions in both PC-9 cells and CIITA-expressing PC-9 cells. **f** Quantification of osimertinib-resistant clones in PC-9 and PC-9-CIITA cells. Data show means of five biological replicates, with error bars representing ±s.e.m; significance was determined by an unpaired, two-tailed Student's *t*-test. **g, h** Detection of CD74-ROS1 fusion by genomic DNA PCR (**g**) and RT-PCR (**h**). Similar results were obtained from *n* = 2 independent experiments. **i** Detection of CD74-ROS1 transcripts in PC-9 and CIITA-expressing PC-9 cells. Similar results were obtained from *n* = 15 resistant clones. Source data are provided as a Source Data file.

Fig. 7f, k, l), most likely due to the presence of >4 copies of the *EGFR* gene in PC-9[24], and frequent breakpoints were observed also in *EPHA2* gene, which is highly transcribed in these cells (Supplementary Fig. 7j, l). All combined, these data suggest that the preferential usage of specific partners or exons during oncogenic TK fusion formation is the result of a selection process among multiple combinations of junctions created by DSBs spontaneously generated in the genome, rather than due to the presence of pre-existing clusters of breakpoints like in the case of AID-initiated translocations.

**Protein stability determines the selection of TK fusion partners**

Next, we investigated the process of TK fusion selection. Consistent with the COSMIC analysis in patients, TK fusion partners obtained by FACTS were mutually exclusive in most cases[29,37–39], with only a few partners shared by multiple TK fusions (Fig. 6a and Supplementary Data 4). Interestingly, some fusion partner genes, such as *TPM3* and *ETV6*, used the same exons when they generated oncogenic fusions with different TKs (Fig. 6b). Thus, we explored functional basis of fusion-partner specificity to each kinase. We

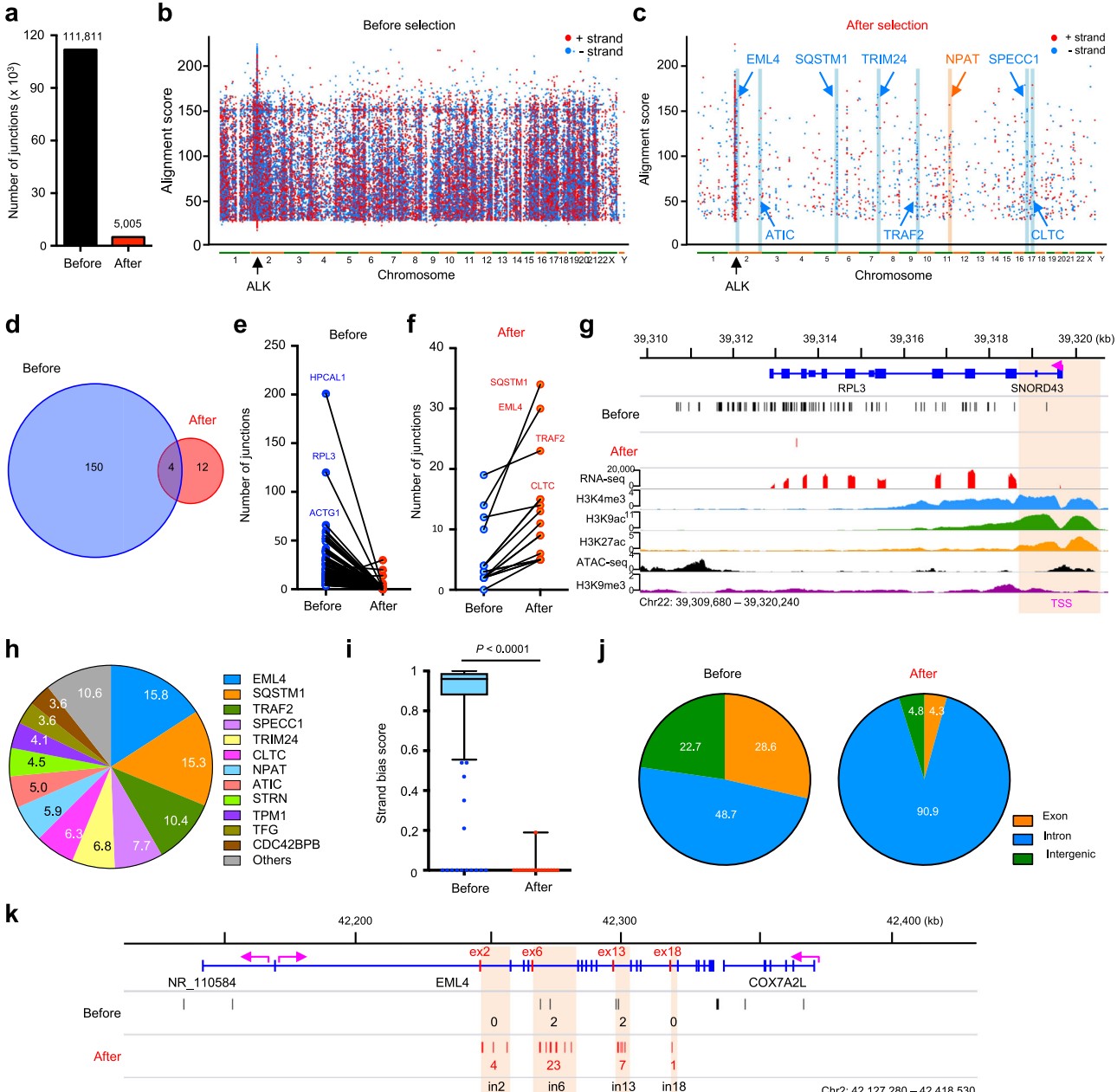

**Fig. 5 | Characterization of functional ALK fusions identified in osimertinib-resistant PC-9 cells by HTGTS. a** The total number of translocations in before and after osimertinib selection by HTGTS. *n* = 3 HTGTS experiment was performed and pooled for DNA breakpoint analysis. **b**, **c** Rainfall plots showing genome-wide distribution of translocations before (**b**) and after (**c**) osimertinib selection in PC-9 cells. Each dot represents a single DNA translocation ordered on the X-axis according to its position in the human genome. Red and blue dots represent the orientation of DNA translocations on chromosome plus or minus strand, respectively. Example genes that transcribe in the plus and minus directions are shown in blue and red, respectively, resulting in functional *ALK* fusions in the correct direction. Data pooled from six biological replicates. **d** Venn diagram showing the overlap of shared DNA translocation hotspots between before and after selection. CRISPR off-targets were excluded. **e**, **f** Change in the number of DNA translocations in 154 hotspots before (**e**) and 16 hotspots after (**f**) osimertinib selection.

**g** Distribution of DNA breakpoints and histone modification marks in *RPL3* gene. Transcription start site (TSS) is indicated in pink. **h** Frequency of ALK translocations with the indicated partner genes. Each color corresponds to each gene. **i** Box plot showing the strand bias score of DNA translocations in 154 hotspots before and 16 hotspots after osimertinib selection. Y-axis represents strand bias score where 1 indicates no strand bias and 0 a 100% strand biased. The minima and maxima of box indicate the 25th to 75th percentile. The centre indicates 50th percentile. The wiskers were drawn down to the 10th percentile and up to the 90th percentile. Significance was determined by an unpaired, two-tailed Student's *t*-test. **j** Pie graphs showing translocation distributions analyzed from hotspots identified before (**left**) and after (**right**) osimertinib selection. **k** Detailed distribution of DNA breakpoints in *EML4* genes. The purple arrows indicate orientation of genes. The number of translocations in focal clusters is indicated in black and red for before and after selection, respectively. Source data are provided as a Source Data file.

engineered all combinations of EML4 and CD74 fusions with ALK, RET, ROS1, and NTRK1 (Supplementary Fig. 8a, b). While all of the fusion junctions were detected equally at the genomic DNA levels (Supplementary Fig. 8c, d), some of the kinase fusion combinations did not yield resistant clones under osimertinib selection

(Fig. 6c, d). While thousands of EML4-ALK, EML4-RET, or EML4-NTRK1 clones rapidly emerged, no clones with EML4-ROS1 fusions were observed (Fig. 6c). Likewise, while thousands of CD74-ROS1 clones emerged, no clones with CD74-ALK, CD74-RET, or CD74-NTRK1 fusions emerged in PC9 cells expressing CIITA (Fig. 6d).

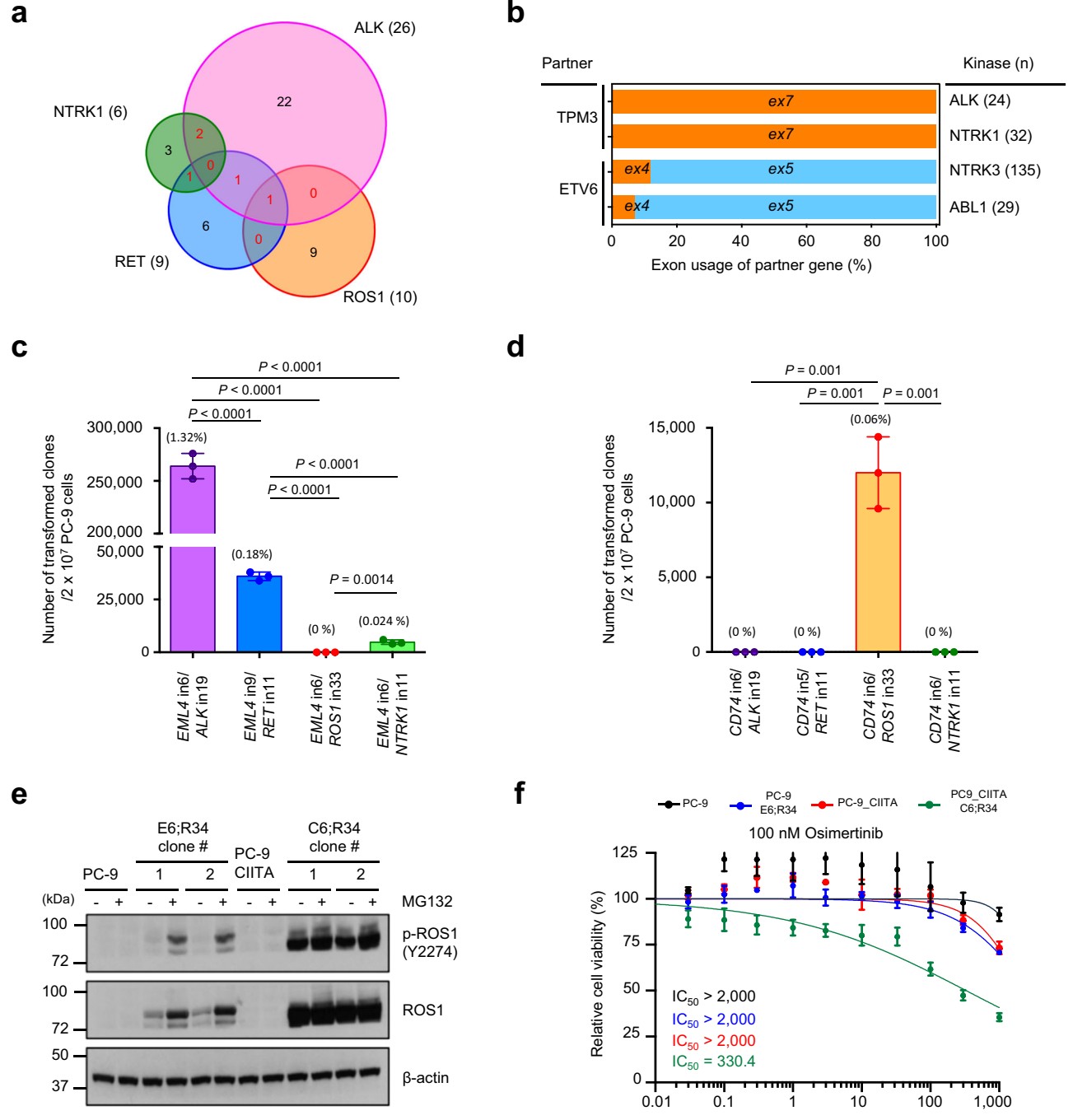

**Fig. 6 | The fusion partners of different TKs are largely exclusive. a** Venn diagrams showing the overlap of fusion partners shared between *ALK* (*n* = 26 partner genes), *RET* (*n* = 9), *ROS1* (*n* = 10), and *NTRK1* (*n* = 6) fusions identified by FACTS. **b** Comparisons of exon usages of partner genes between different kinase genes. **c** Quantification of osimertinib-resistant clones in PC-9 cells with fusions induced by between EML4 and different TKs. Data show means of three biological replicates, with error bars representing ±s.e.m; significance was determined by an unpaired, two-tailed Student's *t*-test. **d** Quantification of osimertinib-resistant clones in PC-9 cells with fusions induced by between CD74 and different TKs. Data show means of three biological replicates, with error bars representing ±s.e.m; significance was determined by an unpaired, two-tailed Student's *t*-test. **e** Representative western blot analysis of EML4-ROS1 and CD74-ROS1 fusions after MG-132 treatment. Similar results were observed in *n* = 2 independent experiments. **f** Sensitivity to combination of osimertinib plus crizotinib in clones with EML4-ROS1 E6; R34 and CD74-ROS1 C6; R34 fusions. Data show means of three biological replicates, with error bars representing ±s.e.m. Source data are provided as a Source Data file.

Next, we isolated single cell-derived clones harboring different fusions for further characterization (Supplementary Fig. 8e). Clones with CD74-ROS1 fusion displayed abundant protein that was phosphorylated as expected, but clones with EML4-ROS1 fusion showed very low abundance of the EML4-ROS1 protein that was also poorly phosphorylated (Fig. 6e). Treatment with proteasome inhibitor MG132 stabilized the EML4-ROS1 fusion protein and its phosphorylation substantially increased (Fig. 6e). Crizotinib, primarily a MET inhibitor with an activity on ROS1, inhibited the growth of clones harboring CD74-ROS1 fusions but not EML4-ROS1 fusions, suggesting that only stable and abundant kinase fusions could create oncogenic dependency (Fig. 6f).

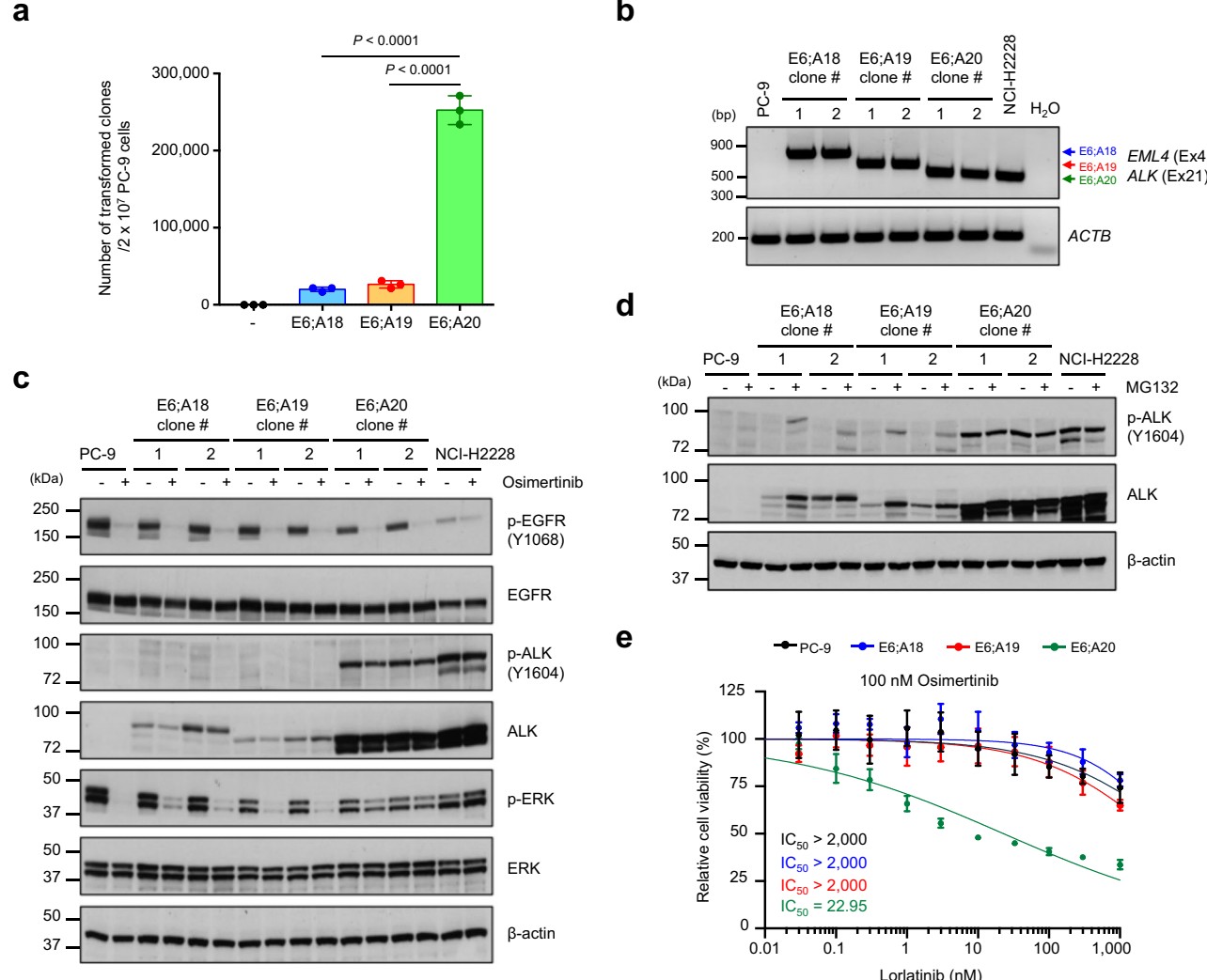

**Fig. 7 | EML4-ALK fusions have the preferential usage of selected exons.**
**a** Quantification of osimertinib-resistant clones in PC-9 cells carrying different EML4-ALK fusion variants. Data show means of three biological replicates, with error bars representing ±s.e.m; significance was determined by an unpaired, two-tailed Student's *t*-test. **b** mRNA expression of EML4-ALK transcripts by RT-PCR. Similar results were obtained from $n = 2$ independent experiments. **c** Representative western blots of the signaling changes in different EML4-ALK

fusion variants treated with EGFR inhibitor (osimertinib). Similar results were observed in $n = 2$ independent experiments. **d** Representative western blot analysis of EML4-ALK fusion variants after MG-132 treatment. Similar results were observed in $n = 2$ independent experiments. **e** Sensitivity to combination of osimertinib plus lorlatinib in clones with different EML4-ALK fusion variants. Data show means of three biological replicates, with error bars representing ±s.e.m. Source data are provided as a Source Data file.

## Protein stability determines the specific exon usage of oncogenic TK fusions

An additional finding of COSMIC analysis was the preferential usage of specific exons in TK fusions (Fig. 1e, f). To better understand molecular basis of preferential exon usage in kinase fusions, we engineered EML4-ALK variants by CRISPR/Cas9 that fuse the same *EML4* exon 6 to different *ALK* exons (e18, e19, or e20) (Supplementary Fig. 8f). All these fusions are predicted to be in frame, which could potentially lead to functional ALK fusions. However, these different fusion variants have been detected in different frequencies in patients, the EML4-ALK E6;A20 fusion being far more frequent than the E6;A18 or E6;A19 fusions (<1% among ALK fusions)[48,49] (Fig. 1e), which was consistently observed in FACTS (Fig. 7a). To understand the cause of these differences, we generated clonal lines for each fusion variant. The mRNA transcription levels were comparable among the variants, likely due to their regulation by the same promoter (Fig. 7b and Supplementary Fig. 8g). However, the protein abundance and the level of phosphorylation were markedly different (Fig. 7c and Supplementary Fig. 8h). The E6;A20 fusion protein was

highly expressed and phosphorylated, whereas the E6;A18 or E6;A19 fusions were much less abundant with barely detectable phosphorylation (Fig. 7c and Supplementary Fig. 8h). Consequently, the E6;A20 variant showed a greater potency in rescuing MAPK pathway activation compared to the E6;A18 or E6;A19 fusions in osimertinib-treated PC-9 cells (Fig. 7c). Treatment with MG132 stabilized the E6;A18 or E6;A19 fusions and led to their phosphorylation (Fig. 7d). Next, we investigated whether the different functional features of these EML4-ALK fusion variants were due to differences in sub-cellular localization, given recent evidence showing that the onco-genic activity of EML4-ALK is dependent on its subcellular localization and formation of protein granules in the cell cytoplasm[50]. The three EML4-ALK fusion variants showed compar-able intracellular localization in confocal microscopy analysis (Sup-plementary Fig. 8i, j), with weaker signals with the E6;A18 or E6;A19 fusions, likely due to their low protein abundance. Functional assay showed that lorlatinib inhibited the growth of cells harboring E6;A20 fusions but not of the E6;A18 and E6;A19 fusions, suggesting that only E6;A20 fusions are stable enough to confer oncogenic dependence

(Fig. 7e). These findings imply that the usage of specific exons in TK fusions is likely dictated by protein stability rather than transcription or subcellular localization of the resulting fusions, and that only an abundant expression of TK fusion proteins creates a dependency that might determine the efficacy of TKI treatment.

**TKI therapy is less effective in patients with atypical ALK fusions**
Since atypical ALK fusions showed reduced functionality and onco-genic signaling in PC-9 cell models (Fig. 7c, e), we investigated whether these findings were reflected in patients by studying clinical responses to ALK TKIs in patients carrying either typical or atypical ALK fusions. We analyzed 108 patients with metastatic NSCLC who tested positive for ALK fusions by next-generation sequencing (NGS) and received ALK TKI treatment and divided them into two groups based on the *ALK* gene fusion breakpoints: typical (breakpoints in *ALK* intron 19) and atypical (breakpoints in other *ALK* introns/exons or atypical fusion partner). There were 97 typical ALK fusions with *ALK* breakpoints in intron 19, and 11 atypical fusions cases with breakpoints in introns 16, 17, 18, and 20 or inside exon 20 (Supplementary Fig. 9a and Supplementary Data 8). Patients with atypical ALK fusions had clinical char-acteristics comparable to patients with typical ALK fusions in terms of age, gender, smoking history, ECOG performance status, and ALK inhibitor treatment (Supplementary Fig. 9b). The typical ALK fusion group had 88.7% (88/97) of EML4-ALK fusions or other known onco-genic ALK fusions, such as HIP1-ALK[51], whereas the group of atypical ALK fusions was composed of 54.5% (6/11) of EML4-ALK fusions with non-intron 19 breakpoints (Supplementary Fig. 9c) or ALK fusions with atypical partners. Strikingly, the atypical group showed significantly lower objective response rate (ORR) to ALK TKI compared to the group of patients with typical ALK fusions (54.5% versus 88.7%, $p = 0.01$) (Supplementary Fig. 9d), resulting in a significantly shorter progression-free survival (PFS; 5 months versus 20.5 months, HR: 0.18 [95% CI: 0.08-0.38], $p < 0.001$) and overall survival (OS; 20.5 months versus 83.0 months, HR: 0.20 [95%CI: 0.09-0.45], $p < 0.001$) (Fig. 8a, b). We also confirmed that atypical ALK fusions retained a significant association with shorter PFS and OS after adjusting for potential con-founders in multivariable Cox regression models (Supplementary Fig. 9e). We further examined co-occurring mutations in cases with typical or atypical ALK fusions. The most frequently mutated gene was *TP53* in typical and atypical ALK fusions, with a significantly higher frequency in atypical fusions (62.5% versus 25.9%, $p = 0.046$), which may have also contributed to the worse outcomes to ALK TKIs observed in this subset of patients (Fig. 8c). In addition, atypical ALK fusions were associated with a higher rate of mutations in alternative oncogenic driver genes, including *BLM*, *FLT4*, *RAF1*, *RB1*, and *TCF3*[52], compared to typical ALK fusions (Fig. 8c and Supplementary Fig. 9f). Overall, these results demonstrate that atypical TK fusions are weaker oncogenic driver, are associated with increased co-mutation of other oncogenes, and respond poorly to ALK inhibition, providing a bio-marker predictor for response to ALK TKI in patients.

## Discussion
In this work we provide mechanistic explanation and clinical relevance of the recurrent patterns of oncogenic TK fusion in cancers. TK fusions show mutually exclusive fusion partners, with just a few partner genes shared by multiple TK genes. In addition, TK and partner genes employ a preferential usage of specific exons. We developed FACTS to study genome-wide the mechanisms of formation of functional chromoso-mal translocations that drive solid tumor growth and resistance to TKI inhibition. Current techniques to map genome-wide chromosomal translocations are mostly focused on early, unbiased mechanistic events of DNA translocation formation largely in B lymphocytes[17,18], without interrogating the oncogenicity of the cloned translocations. In contrast, by FACTS we found that oncogenic ALK, RET, ROS1, and NTRK1 fusions form spontaneously in normal or tumoral lung

epithelial cells when a DSB is introduced in the TK gene. These spon-taneous translocations occur not only in PC-9 lung cancer cells selec-ted in vitro by the pressure of osimertinib, but also in non-tumoral BEAS-2B cells that are transformed in vivo in mice, indicating that FACTS can be applied to tumoral or non-tumoral cells subjected to different selection modalities. Since translocation formation is known to extend and spread for several kilobases flanking the original DSB due to recession of the DNA broken ends during the DNA repair process[53,54], FACTS allows to detect exon fusion variants of TK fusions by designing sequencing primers in multiple exons for fusion detec-tion. By extension, it is conceivable that FACTS could be applied vir-tually to almost any TK gene or any normal or tumoral cells for which a method of selective pressure is available. In principle, the FACTS approach could be possibly expanded to study the formation and selection of other chromosomal rearrangements different from TK fusions occurring in sarcomas, hematologic malignancies, or other tumors, but it needs to be demonstrated by experimental evidence.

By applying FACTS to lung epithelial cells, we discovered key factors leading to the formation of chromosomal translocations in solid tumors. We found that mRNA expression level of the partner gene was essential to the point that reactivation of transcription, such as in the case of *HLA-DRB1*, was sufficient to induce the spontaneous formation of functional fusions (Fig. 4 and Supplementary Fig. 5). Gene transcription was sufficient not only to express the resulting fusion but also to increase the probability of DSBs occurring within introns and exons of a gene with increased transcription, in keeping with the knowledge that transcription levels in a gene correlate with DSB frequency[55,56]. This mechanism has implications for the understanding of the cell of origin in lung cancers driven by chromosomal transloca-tions. For example, lung cancers with HLA-DR or CD74 fusions are most likely to originate in cells that express these genes robustly, such as alveolar type II cells[57,58] (Supplementary Fig. 10a−c).

By comparing patterns of rearrangements before selection by HTGTS to those after selection by FACTS, we further gained insights in the process of oncogenic TK fusion formation. Before selection, HTGTS detected no clusters of DSB breakpoints in TK genes (Supple-mentary Fig. 7b−j) and only few clusters in partner genes (Fig. 5k and Supplementary Fig. 6e−g), suggesting that most TK fusions originate by a selection of translocations that arise from spontaneous DSBs dispersed throughout the genome without pre-determined hotspots. These DSBs are likely generated by various mechanisms, including the formation of R-loops, G4 quadruplex, stalled replication forks, corre-late with active transcription[59] and might be facilitated by enzymatic activity of APOBEC enzymes[60]. Thus, in solid tumors recurrent trans-locations might be selected by different mechanisms than in hema-tologic malignancies, such as B-cell lymphoma, in which recurrent translocation are largely dictated by the off-target activity of the activation-induced cytidine deaminase (AID) and the recombination activating gene (RAG)1/2 enzymes[55,61,62]. Furthermore, recurrent TK translocations do not appear to occur at a high frequency when compared to other translocations, but oncogenic translocations are heavily selected due to their potential to drive cancer cell survival and proliferation.

Comprehensive analysis of transcriptome sequencing data on gene fusions in childhood cancers[63] has shown multiple molecular factors that influence oncogenic gene fusions, including, gene length, splicing translation frame, and protein domains. Our experimental model provided evidence of some of these mechanisms, such as intron-versioning and neo-splicing[63], while also providing evidence for a selection process that enriches for specific TK fusions in human cancers. During the selection process, only genomic breakpoints located in introns with the correct orientation that result in fusion proteins with functional activity were enriched, whereas breakpoints leading to out-of-frame proteins disappeared. Proximity of the genes also played a role because we consistently identified fusions with genes

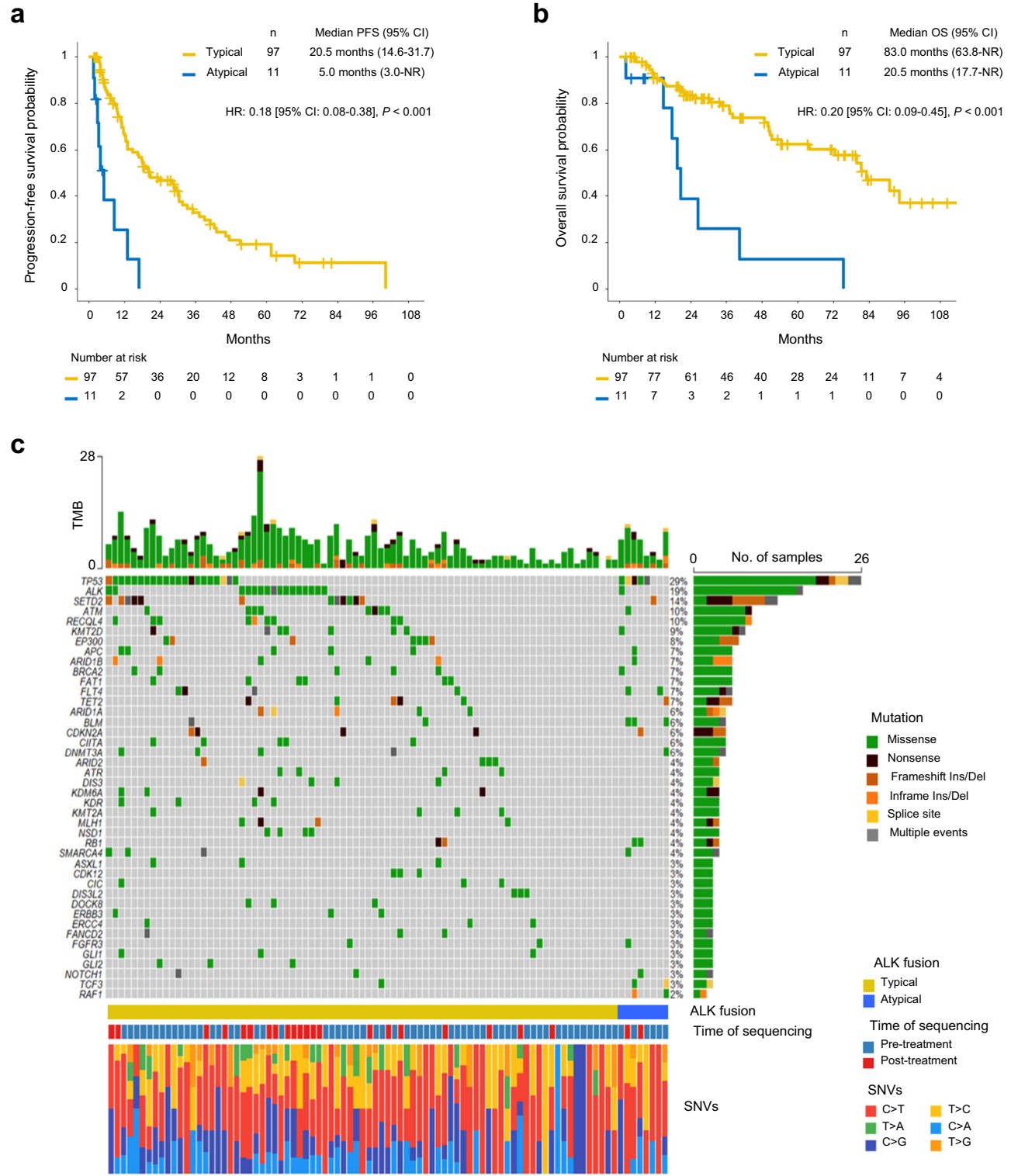

**Fig. 8 | Atypical ALK fusions show a poor survival compared to typical ALK fusions. a**, **b** Progression-free (**a**) and overall (**b**) survival in patients carrying typical and atypical ALK fusions treated with ALK inhibitors. *P*-values were determined by two-sided, and confidence intervals were at the 95% level. **c** OncoPrint plot showing the top 42 genes mutated among NSCLC harboring typical and atypical ALK fusions. TMB tumor mutational burden, SNVs single nucleotide variants.

located on the same chromosome that constitute a topological domain with a higher probability of contact[46,47,64]. While the generation of in-frame and highly transcribed fusions with a partner that have dimerization domains are expected mechanistic factors, they are insufficient to explain the specific exon usage or partner usage of each TK fusion as it is observed in patients. For example, EML4 is the most frequent fusion partner for ALK, but it is very rarely translocated with RET, ROS1 or NTRK1; likewise, CCDC6-RET and CD74-ROS1 fusions are frequently found in NSCLC patients, but CCDC6 fusion with ALK or ROS1 and CD74 fusion with ALK have never been described[29,37–39] (Supplementary Fig. 10d–g). We discovered that protein stability is a key determinant factor that could be not predicted simply based on the characteristic of the fusion partner. By investigating EML4-ALK fusions with different *ALK* exons, we demonstrated that a specific exon

usage was critical to provide stability to the fusion protein (Fig. 7). We found that inhibition of the proteasome by MG-132 in the unstable EML4-ALK E6;A18 and E6;A19 variants increased protein abundance (Fig. 7d). Recent study suggested that degrons, short motifs that affect protein degradation rate (i.e. D box, KEN box, SPOP motif, and PEST sequences), could regulate the expression of fusion proteins by the degron loss mechanism[65]. PEST sequences[66] are present in the unstable EML4-ALK E6;A18 and E6;A19 variants but not in the stable oncogenic EML4-ALK E6;A20 variant, implying that degron gain in the EML4-ALK E6;A18 and E6;A19 variants might contribute to their rapid degradation through the ubiquitin-proteasome pathway. Similarly, we observed that oncogenic fusions were stable only when each TK portion was paired with a specific partner but with other partners (Fig. 6). While the CD74-ROS1 fusion was stable and phosphorylated, the EML4-ROS1 fusion, which is not seen in patients, was unstable and poorly phosphorylated (Fig. 6e). Thus, the selection process of the specific fusion partner for each TK depends not only on the availability of an in-frame dimerization domain but also on the stability of the resulting fusion. Furthermore, the pattern of recurrent oncogenic TK fusions may be dictated by the protein stability of the resulting fusion proteins rather than the enriched DSBs in specific locations in the cancer genome. Yet, we cannot exclude that other mechanisms other than protein stability could contribute to the selection of the oncogenic TK fusions in cancers.

Accurate detection of oncogenic TK fusions is critical to effective treatment for cancer patients with TK fusions. Several methods, such as fluorescence in situ hybridization (FISH), immunohistochemistry (IHC), quantitative real-time PCR (qRT-PCR), targeted DNA sequencing, and targeted RNA sequencing, are routinely employed to diagnose TK fusions. Although these techniques are sufficient for detecting TK fusions, they do not provide functional evidence for their oncogenic activity. Indeed, recent studies showed that some patients do not respond to ALK TKIs despite IHC and NGS confirmation of ALK fusions[67,68]. We demonstrate here that the stability and functionality of the TK fusion proteins is a key factor for the TKI response (Figs. 6 and 7), indicating that the detection of TK fusion junctions by DNA- and RNA-based sequencing does not fully predict the response to TKI therapy. We showed that patients with atypical EML4-ALK rearrangements encoding for unstable fusion proteins responded poorly to ALK TKIs and had shorter PFS and OS compared to patients with typical EML4-ALK (Fig. 8a, b and Supplementary Fig. 9d). Because patients with atypical EML4-ALK fusions showed partial and transient response to ALK targeted therapy (Supplementary Fig. 9d and Supplementary Data 8), these atypical fusions may be very weak oncogenic drivers. Although these data are still limited by the small sample size, they suggest that a strong oncogenic dependency by tumor cells develops only when the TK fusion is stably expressed, and that this dependency predicts TKI response in patients. Thus, studies on novel ALK inhibitors should stratify results by TK junction as patients with atypical ALK junctions may not respond as well.

## Methods

### Cell lines and cultures

PC-9 cells were a kind gift from Dr. Pasi A. Jänne (Dana-Farber Cancer Institute, Boston, MA, catalog number: 90071810; Millipore Sigma) and were maintained in RPMI 1640 medium (catalog number: 15-040-CV; Corning) supplemented with 10% fetal bovine serum (FBS, catalog number: 10437-028; Gibco), 100 units/ml penicillin-streptomycin (catalog number: 30-002-CI; Corning), and 2 mM L-Glutamine (catalog number: 25-005-CI; Corning). To select osimertinib-resistant transformed PC-9 clones, cells were transduced with CRISPR/Cas9-containing lentiviral supernatants targeting *EML4* intron 6, *EML4* intron 9, *EML4* intron 13, *ALK* intron 17, *ALK* intron 18, *ALK* intron 19, *RET* intron 11, *ROS1* intron 33, *NTRK1* intron 11, *CD74* intron 5, or *CD74* intron 6 with different combinations in the presence of 6 µg/ml

polybrene (catalog number: sc-134220; Santa Cruz Biotechnology). The viral supernatants were replaced with fresh culture medium 6 h later. Next day, cells were plated in 12-well plates (4 ×10⁴ cells/well) and 10 cm dishes (5 × 10⁶ cells/dish). Following day, cells were selected with 100 ng/ml of osimertinib (AZD9291, EGFR inhibitor, catalog number: S7297; Selleckchem) every other day for 30 days. Frequency of resistant clones was calculated by counting the resistant clones/20 million cells. Transformed clones were collected for functionally active chromosomal translocation sequencing (FACTS) and HTGTS to use as the samples for after osimertinib selection. For FACTS and HTGTS experiments for the samples of after osimertinib selection (bait in intron 19 of *ALK*), we pooled samples of osimertinib resistant clones both in PC-9 cells and in PC-9 cells overexpressing apolipoprotein B mRNA editing enzyme, catalytic polypeptide-like 3 C (APOBEC3C). For HTGTS experiment for the samples of before osimertinib selection, PC-9 cells were transduced with CRISPR/Cas9-containing lentiviral supernatants targeting *EML4* intron 6 or *ALK* intron 19 in the presence of 6 µg/ml polybrene. Cells were collected after 5 days of DNA double-strand breaks (DSBs) in *ALK* intron 19.

To generate PC-9 clones carrying EML4 (Ex6)-ALK (Ex18), EML4 (Ex6)-ALK (Ex19), EML4 (Ex6)-ALK (Ex20), EML4 (Ex6)-ROS1 (Ex34), or CD74 (Ex6)-ROS1 (Ex34) fusion, PC-9 cells were transduced with different combinations of lentiviral particles and seeded in 96-well plates for single cell cloning. PC-9 cells expressing the fusion protein were validated by RT-PCR.

To generate PC-9 cells expressing CIITA, PC-9 cells were transfected with pcDNA3 myc CIITA (P#808) plasmid (plasmid #14650; Addgene), were selected with 0.5 mg/ml of geneticin (G418 sulfate, Gibco) by replacing geneticin every other day for 2 weeks and were sorted by staining with APC-conjugated HLA-DR antibody by BD FACSAria II cell sorter (BD Bioscience). Stable PC-9 cells expressing CIITA were maintained in the medium containing 0.5 mg/ml geneticin.

BEAS-2B (catalog number: CRL-3588) were obtained from ATCC and cultured in Airway Epithelial Cell Basal Medium (catalog number: PCS-300-030; ATCC) with Bronchial Epithelial Cell Growth Kit (catalog number: PCS-300-040; ATCC) containing 500 µg/ml HAS, 0.6 µM Linoleic Acid, 0.6 µg/ml Lecithin, 6 mM L-glutamine, 0.4% Extract P, 1 µM Epinephrine, 5 µg/ml Transferrin, 10 nM T3, 1 µg/ml Hydrocortisone, 5 ng/ml recombinant Human EGF, and 5 µg/ml recombinant Human Insulin. Cells were transduced with CRISPR/Cas9-containing lentiviral supernatants targeting *EML4* intron 6 or *ALK* intron 19 with 6 µg/ml polybrene. As positive controls, cells were transduced with CRISPR/Cas9-containing lentiviral supernatants targeting both *EML4* intron 6 or *EML4* intron 13 and *ALK* intron 19 with 6 µg/ml polybrene. The viral supernatants were replaced with fresh culture medium 6 h later. Two days later, cells were collected for mouse xenograft experiments.

293FT (catalog number: R70007; ThermoFisher Scientific) cells were maintained in DMED medium (catalog number: 15-017-CV; Corning) supplemented with 10% FBS, 100 units/ml penicillin-streptomycin, and 2 mM L-Glutamine. HEK293FT cells were used to produce CRISPR/Cas9-containing lentiviral particles.

NCI-H2228 (catalog number: CRL-5935; ATCC) and NCI-H3122 (catalog number: 300484; Cytion) were maintained in RPMI 1640 medium supplemented with 10% FBS, 100 units/ml penicillin-streptomycin, and 2 mM L-Glutamine. NCI-H2228 and NCI-H3122 were used as positive controls for EML4-ALK (E6-A20; variant 3a/b) and EML4-ALK (E13-A20; variant 1), respectively.

All cell lines were tested negative for *mycoplasma* contamination and were cultured at 37 °C in 5% CO₂ atmosphere.

### Lentiviral particle productions

To produce lentivirus, 5.5 × 10⁶ 293FT cells were plated in a 10 cm dish day before transfection. The following day, cells were transfected using Xfect transfection reagent (catalog number: 631318; Takara Bio)

with 20 µg of lentiCRISPR/Cas9 plasmid, 3.6 µg of pMD2.G (Addgene plasmid #12259), 3.6 µg of pRSV-Rev (Addgene plasmid #12253) and 3.6 µg of pMDLg/pRRE (Addgene plasmid #12251)[69]. The medium was changed with complete culture medium 6 h post-transfection. The viral supernatant was collected 48 h post-transfection, passed through a 0.45-µm syringe filter (PVDF membrane, catalog number: 89414-902; VWR), pooled, and used either fresh or snap frozen.

## EGFR, ALK, RET, and proteasome inhibitors

Osimertinib (AZD9291, EGFR inhibitor, catalog number: S7297) was purchased from Selleckchem. To select osimertinib resistant PC-9 clones, we used 100 nM osimertinib. For cell proliferation assay, osimertinib was used at 0.03 nM to 1000 nM concentrations as indicated in the corresponding figure legend. For western blot experiments, cells were treated with 100 nM of osimertinib for 4 h.

Lorlatinib (PF-6463922, ALK inhibitor) was obtained from Pfizer. For cell proliferation assay, lorlatinib was used at 0.03 nM to 1000 nM concentrations as indicated in the corresponding figure legend. For western blot experiments, cells were treated with100 nM of lorlatinib for 4 h.

Selpercatinib (LOXO-292, RET inhibitor, catalog number: S8781) was purchased from Selleckchem. For cell proliferation assay, selpercatinib was used at 0.03 nM to 1000 nM concentrations as indicated in the corresponding figure legend.

Crizotinib was obtained from Pfizer. For cell proliferation assay, crizotinib was used at 0.03 nM to 1000 nM concentrations as indicated in the corresponding figure legend.

MG-132, a potent, reversible, and cell-permeable proteasome inhibitor, was purchased from Sigma-Aldrich (catalog number: 474790). For proteasome inhibition experiments, cells were treated with 2 µM of MG-132 for 18 h.

## Cell proliferation assay

PC9 cells ($4 \times 10^3$ cells/well) were plated into 96-well plates and were treated with either EGFR inhibitor (osimertinib), ALK inhibitor (lorlatinib), RET inhibitor (selpercatinib), or ROS1 inhibitor (crizotinib) as well as combination with both EGFR inhibitor and either ALK inhibitor or RET inhibitor or ROS1 inhibitor for 72 h. 20 µl of CellTiter 96® Aqueous One Solution Reagent (catalog number: G3582; Promega) was added into each well and cells were incubated for 1 h. The absorbance was measured at 490 nm using a 96-well plate reader. Inhibitory concentration ($IC_{50}$) values were derived by a sigmoidal dose-response (variable slope) curve using GraphPad Prism software.

## Clonogenic assay

PC9 cells ($4 \times 10^4$ cells/well) were seeded in 24-well plates day before the experiments. Cells were induced DNA DSBs in both EML4 (either intron 6 or intron 13) and ALK (intron 19). Two days later, cells were treated with 100 nM of osimertinib every other day for 7 days. At day 7, cells were washed with ice-cold PBS two times and fixed with ice-cold methanol for 10 min on ice. Methanol was aspirated and 1 ml of 0.5% crystal violet solution (made in 25% methanol) was added in each well. After 10 min of incubation at room temperature, crystal violet solution was removed, and plates were carefully rinsed in water until color is no longer coming off in rinse. Plates were dried at room temperature and were scanned.

## CRISPR/Cas9 sgRNA design and cloning

For SpCas9 expression and generation of single guide RNA (sgRNA), the 20-nt target sequences were selected to precede a 5'-NGG protospacer-adjacent motif (PAM) sequence. The human EML4-intron 6-targeting sgRNA, human EML4-intron 9-targeting sgRNA, the human EML4-intron 13-targeting sgRNA, the human ALK-intron 17-targeting sgRNA, the human ALK-intron 18-targeting sgRNA, the human ALK-intron 19-targeting sgRNA, the human RET-intron 11-targeting sgRNA,

the human ROS1-intron 33-targeting sgRNA, the human NTRK1-intron 11-targeting sgRNA, human CD74-intron 5-targeting sgRNA, and human CD74-intron 6-targeting sgRNA were designed with the CRISPR design tool (CRISPick; https://portals.broadinstitute.org/gppx/crispick/public). Oligonucleotides synthesized by Integrated DNA technology (IDT) were annealed and cloned into the BsmbI-BsmbI sites downstream from the human U6 promoter in LentiCRISPR v2 plasmid (Addgene plasmid #52961). sgRNA sequences were confirmed by Sanger sequencing with U6 promoter primer 5'-GAGGGCCTATTTCCCATGAT-3'. Oligonucleotides for sgRNA cloning used in the study are listed in Supplementary Data 9.

## Mice and tumor xenograft models

Animal experiments were performed under the mouse protocol approved by Institutional Animal Care and Use Committee (IACUC) of Boston Children's Hospital (protocol #00001530). All mice were housed and maintained in individually ventilated cages with a 12-h light-dark cycle and with ad libitum access to food and water in the specific pathogen free (SPF) facility at Boston Children's Hospital. Mice were housed in temperatures of 18–24 °C with 40-60% humidity. Immunodeficient NOD SCID gamma (NSG) mice (NOD.Cg-Prkdc$^{scid}$ Il2rg$^{tm1Wjl}$/SzJ, Stock number: 005557; The Jackson Laboratory) were used for xenograft experiments. We used both male and female mice at the ages of 6–12 weeks. BEAS-2B cells ($5 \times 10^6$ cells/mouse) transduced with the CRISPR/Cas9 system targeting ALK intron 19 were subcutaneously injected in both flanks of twenty NSG mice. As positive controls, BEAS-2B cells ($5 \times 10^6$ cells/mouse) transduced with the CRISPR/Cas9 system targeting both ALK intron 19 and either EML4 intron 6 or EML4 intron 13 were subcutaneously injected in both flank of two NSG mice. Tumor growth was measure with a caliper every seven days. No mice were excluded from the analysis and no randomization or blinding method was used. The maximal tumor size permitted was 1.5 cm in maximum diameter for tumors. The maximal tumor size was never exceeded.

## Genomic DNA isolation

Genomic DNA (gDNA) was extracted from PC-9 cells and osimertinib-resistant transformed PC-9 clones using rapid lysis buffer (100 mM Tris-HCl pH8.0, 200 mM NaCl, 5 mM EDTA, 0.2% SDS) containing 10 µg/ml Proteinase K (catalog number: P2308; Sigma Aldrich). After overnight incubation at 56 °C, gDNA was precipitated in one volume isopropanol, and the DNA pellet was resuspended in Tris-EDTA (TE) buffer. gDNA was used for HTGTS library preparation.

## RNA extraction, RT-PCR and qRT-PCR

Total RNA was isolated from the cells using RNeasy Plus Mini Kit (catalog number: 74136; Qiagen) following the manufacturer's instructions. cDNA was synthesized using iScript cDNA synthesis kit (catalog number: 1708891; Bio-Rad) following the manufacturer's instructions. All qRT-PCR experiments were performed in triplicate on ICycler iQ Real-Time PCR Detection System (Bio-Rad) with iTaq universal SYBR green supermix (catalog number: 1725121, Bio-Rad). Expression levels for individual transcripts were normalized against β-actin. Fold change in transcript levels were calculated as fold change over H2228 cells. Primers for RT-PCR and qRT-PCR are listed in Supplementary Data 9.

## Flow cytometry

To check surface expression of HLA-DR and CD74, cells were washed with ice-cold PBS, were stained with APC-conjugated HLA-DR antibody (clone: C243, catalog number: 340549, BD Biosciences, 1:100) and PE-conjugated CD74 antibody (clone: LN2, catalog number: 326808, BioLegend, 1:100) for 30 min on ice, were analyzed using a BD FACS-Celesta flow cytometer (BD Biosciences). For intracellular staining, cells were washed with ice-cold PBS, were fixed/permeablized with

Fixation/Permeabilization kit (catalog number: 554714, BD Bsiosciences), stained with APC-conjugated HLA-DR antibody (clone: C243, catalog number: 340549, BD Biosciences, 1:100) and PE-conjugated CD74 antibody (cytoplasmic antibody, clone: Pin.1, catalog number: 357604, BioLegend, 1:100) for 30 min on ice, were analyzed using a BD FACSCelesta flow cytometer (BD Biosciences). Data were analyzed by FlowJo software (FlowJo).

## Protein extraction and western blot analysis
Whole-cell extracts were obtained from PC-9 cells, BEAS-2B cells, NCI-H2228 cells, and NCI-H3122 cells using GST buffer (10 mM MgCl$_2$, 150 mM NaCl, 1% NP-40, 2% Glycerol, 1 mM EDTA, 25 mM HEPES (pH7.5)) or NP-40 buffer (50 mM Tris-HCl (pH 8.0), 150 mM NaCl, 1% NP-40) supplemented with protease inhibitors (catalog number: 11697498001; Sigma Aldrich), 1 mM phenylmethanesulfonylfluoride (PMSF), 10 mM NaF, and 1 mM Na$_3$VO$_4$. Extracts were cleared by centrifugation at 20,000 x g for 20 min. The supernatants were collected and assayed for protein concentration using the BCA method (catalog number: BCA1-1KT; Sigma Aldrich). Proteins (30 μg) were loaded on 4–15% Mini-Protean TGX Precast Protein Gels (catalog number: 4561085; BIO-RAD) or 4-12% Criterion XT Bis-Tris Protein Gel (catalog number: 3450124; BIO-RAD), transferred on nitrocellulose membrane or PVDF membrane (GE Healthcare), and blocked with 5% nonfat dry milk (catalog number: 1706404; BIO-RAD). Primary antibodies for immunoblotting included: ALK (clone: D5F3, catalog number: 3633 S; Cell Signaling Technology, 1:2000), phospho-ALK (Tyr1604, catalog number: 3341 S; Cell Signaling Technology, 1:1000), EGF receptor (catalog number: 2232 S; Cell Signaling Technology, 1:1000), phospho-EGF receptor (Tyr1068, clone: D7A5; catalog number: 3777 S; Cell Signaling Technology, 1:1000), p44/42 MAPK (Erk1/2) (clone: 137F5, catalog number: 4695 S; Cell Signaling Technology, 1:1000), and phospho-p44/42 MAPK (Erk1/2) (Thr202/Tyr204, clone: D13.14.4E, catalog number: 4370 S; Cell Signaling Technology, 1:1000), ROS1 (clone: D4D6, catalog number: 3287; Cell Signaling Technology, 1:2000), phospho-ROS1 (Tyr2274; Cell Signaling Technology, 1:1000), and β-actin (clone: 13E5, catalog number: 5125 S; Cell Signaling Technology, 1:2000). Membranes were developed with ECL solution (catalog number: RPN2232; GE Healthcare).

## Membrane protein extraction
Membrane proteins were extracted using Mem-PER Plus Membrane Protein Extraction Kit (catalog number: 89842; ThermoFisher Scientific) by manufacture's instructions. Cells were harvested and washed with cell wash solution. Cytosolic proteins were extracted with permeabliization buffer and the permeabilized pellets were treated with solubilization buffer to extract solubilized membrane and membrane-associated proteins. Cytosolic and membrane fractions were immediately used for western blot analysis.

## Immunohistochemistry and histology
For histology, tumor tissues were fixed with 10% neutral buffered formalin (catalog number: HT501128; Sigma Aldrich) at room temperature for 18 h and washed with 70% ethanol for 24 h. The formalin-fixed tissues were embedded in paraffin blocks and the paraffin-embedded tissue blocks were cut at 4 μm thickness. For immunohistochemistry, formalin-fixed sections were de-waxed in xylene and dehydrated by passage through graded alcohols to water; sections were microwaved in citrate buffer (pH6.0) for 15 min and then transferred to PBS. Endogenous peroxidase activity was blocked by incubating sections in 1.6% hydrogen peroxide solution for 10 min at room temperature. Sections were rinsed in distilled water. Nonspecific staining was blocked with blocking buffer (1% bovine serum albumin (BSA) in PBS) for 1 h at room temperature. The slides were incubated with anti-ALK (clone: D5F3, catalog number: 3633 S; Cell Signaling Technology, 1:100) for 1 h at room temperature. After washing, sections were incubated with biotinylated secondary goat antibody to rabbit IgG and visualized with the Dako EnVision System (code number: K4003; Agilent). The sections were also stained with hematoxylin and erosin (H&E).

## Immunofluorescence
Cells ($6 × 10^4$ cells/well) were seeded on Lab-Tek 8-well chamber slide (catalog number: 154534; ThermoFisher Scientific). The following day, cells were fixed with 4% paraformaldehyde solution (catalog number: J19943; ThermoFisher Scientific) for 20 min, washed, permeabilized with 0.1% Tween-20 (catalog number: P9416; Sigma-Aldrich) for 20 min, and blocked with 5% normal goat serum (catalog number: ab7481; Abcam) for 1 h at room temperature. Cells were incubated with both anti-Golgin-97 (clone: D8P2K, catalog number: 13192 S; Cell Signaling Technology, 1:200) or anti-EEA1 (clone: C45B10, catalog number: 3288 S; Cell Signaling Technology, 1:200) and anti-ALK (clone: 4C5B8, catalog number: 35-4300; ThermoFisher Scientific, 1:200) overnight in the dark at 4 °C. After washing out the primary antibodies, cells were incubated with Alexa fluor 568-conjugated goat anti-rabbit antibody (catalog number: A-101011; ThermoFisher Scientific, 1:100) and Alexa fluor-488 conjugated anti-mouse antibody (catalog number: A32723; ThermoFisher Scientific, 1:100) for 1 h in the dark at room temperature. Cells were washed and mounted using Duolkin In Situ mounting medium with DAPI (catalog number: DUO82040; Sigma-Aldrich). Slides were analyzed using a TCS SP8-STED microscope (Leica) using a 100X objective. Images were analyzed using Image J software.

## Facts
PC-9 (for in vitro model) and BEAS-2B (for in vivo model) cells were introduced DNA DSBs in intron 19 of *ALK*, intron 11 of *RET*, intron 33 of *ROS1*, and intron 11 of *NTRK1* by CRISPR/Cas9. Cells were selected with EGFR inhibitor osimertinib (100 nM) for 30 days by changing the media every other day or in NSG mice until tumor is formed. Fusions were detected by anchored multiplex PCR[28] followed by sequencing and analysis[70]. Total nucleic acid (TNA) was extracted from cell pellets using a Promega Maxwell RSC instrument with a Promega Maxwell RSC SimplyRNA Kit (Madison WI). Libraries were prepared using a Boston Children's Hospital custom assay from ArcherDX (Boulder, CO) whose targets include *ALK*, *ROS1*, *RET*, and *NTRK1*. Several primers were used to target the downstream exons of the targeted introns by CRISPR/Cas9. Libraries were sequenced on Illumina MiSeq instruments (San Diego, CA), and analyzed using Archer Analysis v5.0.6 software from ArcherDX and manually checking frame and protein domains.

A total 155 ALK fusion-driven osimertinib resistant clones were sequenced by 8 independent 3′ end directed fusion assays. A total 60 RET fusion-driven osimertinib resistant clones were sequenced by 2 independent 3′ end directed fusion assays. A total 33 ROS1 fusion-driven osimertinib resistant clones were sequenced by one 3′ end directed fusion assay. A total 22 NTRK1 fusion-driven osimertinib resistant clones were sequenced by one 3′ end directed fusion assay. A total 6 ALK fusion-driven tumors were sequenced by one 3′ end directed fusion assay. A total 70 ROS1 fusion-driven osimertinib resistant clones in CIITA-expressing PC-9 cells were sequenced by 2 independent 3′ end directed fusion assays.

## Generation of HTGTS libraries
HTGTS libraries were generated by emulsion-mediated PCR (EM-PCR) methods[1]. gDNA was digested with HaeIII enzyme (catalog number: R0108; New England Biolabs) overnight. HaeIII-digested blunt ends were A-tailed with Klenow fragment (3′- > 5′ exo-, catalog number: M0212; New England Biolabs). An asymmetric adaptor (composed of an upper liner and a lower 3′-modified linker; Supplementary Data 9) was then ligated to fragmented DNA. To remove the unrearranged endogenous *ALK* and *EML4* locus, ligation reactions were digested with

PvuII (catalog number: R0151L; New England BioLabs) and BbvCI (catalog number: R0601L; New England BioLabs) for *ALK* locus and KpnI (catalog number: R0142L; New England BioLabs) for *EML4* locus, respectively. In the first round of PCR, DNA was amplified using an adaptor-specific forward primer and a biotinylated reverse ALK primer oriented to capture the 5' portion of ALK junction and using a biotinylated forward EML4 primer and an adaptor-specific reverse primer with Phusion High-Fidelity DNA polymerase (catalog number: F530S; Thermo Fisher Scientific). Twenty cycles of PCR were performed in the following conditions: 98 °C for 10 s, 58 °C for 30 s, and 72 °C for 30 s. Biotinylated PCR products were enriched using the Dynabeads MyOne streptavidin C1 (catalog number: 65002; Thermo Fisher Scientific), followed by an additional digestion with blocking enzymes for 2 h. Biotinylated PCR products were eluted from the beads by 30 min incubation with 95% formamide/10 mM EDTA at 65 °C, and purified using Gel Extraction Kit (catalog number: 2870; Qiagen). In the second round of PCR, the purified products were amplified with EM-PCR in an oil-surfactant mixture. The emulsion mixture was divided into individual aliquots and PCR was performed using the following conditions: 20 cycles of 94 °C for 30 s, 60 °C for 30 s, and 72 °C for 1 min. The PCR products were pooled and centrifuged for 5 min at 15,000 x g to separate the PCR product-containing phase and the oil layer. The layer was removed, and the PCR products were extracted with diethyl ether three times. EM-PCR amplicons were purified using the Gel Extraction Kit. The third round of PCR (10 cycles) was performed with the same primers as in the second round of PCR, but with the addition of linkers and barcodes for Illumina Mi-seq sequencing. The third round PCR products were size-fractionated for DNA fragments between 300 and 1000 base pairs on a 1% agarose gel (catalog number: 1613102; Bio-Rad). The PCR products containing Illumina barcodes were extracted with the Gel Extraction Kit.

Nucleotide sequences of junctions were generated by Mi-seq (Illumina NS500 PE250) sequencing at the Molecular Biology Core Facility of the Dana-Farber Cancer Institute. At least three independent libraries were generated and analyzed for each experimental condition. Oligonucleotide primers used for *ALK* library preparations are listed in Supplementary Data 9.

### HTGTS data analysis

**Data process and alignment.** HTGTS sequencing data were processed and aligned[54]. Reads for each experimental condition were demultiplexed by designed barcodes. To enhance specificity and ensure analyzed sequences containing bait portion, reads were further filtered by the presence of primer plus additional 5 downstream bases. The barcode, primer and bait portion of the remained sequences were masked for alignment analysis.

Next, the processed sequences were aligned to human genome (GRCh38/hg38) using BLAT, and finally aligned sequences were further cleaned by removing PCR repeats (reads with same junction position in alignment to the reference genome and a start position in the read <3 bp apart), invalid alignments (including alignment scores <30, reads with multiple alignments having a score difference <4 and alignments having 10-nucleotide gaps), and ligation artifacts (for example, random HaeIII restriction sites ligated to bait breaksite). Translocation junction position was determined based on the genomic position of the 5' end of the aligned read.

**Hotspot identification.** Based on translocation junctions pooled from biological replicates, we identified HTGTS hotspots. First, we detected candidate hotspots where HTGTS junctions were significantly enriched against global genome-wide background by employing SICER 2[71]. The parameters were as follows: window size, 500; gap size, 2000; fdr, 0.01; e value, 10; redundancy, 1; effective genome fraction, 0.74 for human. Next, we eliminated the following candidates: (1) in the region ± 2 Mb around bait ALK break site; (2) with the junction number <5; (3)

without significant local enrichment. The local background *p*-value was calculated by Poisson distribution against the region that surrounds the hotspot (±5 times the size of the hotspot). Bonferroni correction was used to adjust *p*-value for multiple tests. We set adjusted $P = 0.01$ as significance level. The hotspot clusters in the same gene were merged as one.

**Hotspot junctions strand bias score.** We used the following entropy formula to score strand bias as $S = -P \times \log_2(P) - (1 - P) \times \log_2(1 - P)$, where P is the percentage of junctions from the plus stand, and 1-P is the percentage of junctions from the minus strand.

**Alignment score.** Alignment score is BLAT SCORE generated by BLAT tool and calculated by the number of matches with a penalty for mismatches and gaps.

### COSMIC gene fusion analysis

Kinase gene fusions curated from COSMIC v98 (released on May 23, 2023; https://cancer.sanger.ac.uk/cosmic) were collected and manually analyzed.

### ATAC-seq and ChIP-seq data analysis

ATAC-seq and ChIP-seq reads are aligned against to human genome (hg38) using BWA[72]. Aligned BAM files are filtered by removing low quality and unpaired reads, and de-duplicated. Based on the processed BAM files, peak calling is performed using MACS2 peak caller[73]. PC-9 data were acquired from the public databases, including NCBI Gene Expression Omnibus (GEO) and Encyclopedia of DNA Elements (ENCODE). (ATAC-seq: GSM1904729, H3K4me3: ENCSR441JWF, H3K9ac: GSM2534660, H3K9me3: GSM1912806, H3K27ac: ENCSR769FOC).

### Gene expression analysis

RNA-seq data for PC-9 cells were acquired from GEO database (accession number: GSM4635290). Raw reads were aligned to hg38 using STAR v2.6 and FPKM was calculated as expression level. Gene expression data for lung cancer cell lines were acquired from cancer genome project (CGP) accessible at http://www.cancerrxgene.org. Gene expression profiling was measured by affymetrix Human Genome U219 array. The raw expression data were normalized and log$_2$ transformed using RMA method.

### Analyses for patients with ALK fusion-positive NSCLC

Clinicopathologic, genomic, and outcomes data were collected from patients with NSCLC who had consented to institutional review board-approved correlative research protocols Dana-Farber/Harvard Cancer Center (DF/HCC) 02-180, 11-104, 13-364, and/or 17-000 at the Dana-Farber Cancer Institute (DFCI), and whose tumors underwent comprehensive tumor genomic profiling at the DFCI. We screened 300 patients and identified 97 patients with full comprehensive genomic profiling, stage IV, who had received at least one dose of ALK TKIs. Of these, we collected 11 atypical fusions.

**Tumor genomic profiling using OncoPanel.** Targeted next-generation sequencing was performed using the validated OncoPanel assay in the Center for Cancer Genome Discovery at DFCI for 277 (POPv1), 302 (POPv2), or 447 (POPv3) cancer-associated genes[74]. Sequence reads were aligned to reference sequence b37 edition from the Human Genome Reference Consortium using bwa (http://biobwa.sourceforge.net/bwa.shtml), and further processed using Picard (v1.90, http://broadinstitute.github.io/picard/) to remove duplicates and Genome Analysis Toolkit (GATK) to perform localized realignment around indel sites. Single nucleotide variants were called using MuTect v1.1.4, insertions and deletions were called using GATK Indelocator, and variants were annotated using Oncotator. In the DFCI cohort, to

filter out potential germline variants, the standard pipeline removed SNPs present at >0.1% in Exome Variant Server, the NHLBI GO Exome Sequencing Project (ESP) (http://evs.gs.washington.edu/EVS/), present in dbSNP, or present in an in-house panel of normals, but rescues those also present in the COSMIC database. For this study, variants were further filtered by removing variants present at >0.1% in the gnomAD v.2.1.1 database or were annotated as Benign or Likely Benign in the ClinVar database[75,76]. Graphical representation of the most commonly mutated genes in groups of interests and gene enrichment analysis and was performed using the R package maftools[76]. Genomic analysis was performed in the subset of patients with ALK-positive NSCLC treated with ALK kinase inhibitors who had in-house full genomic annotation available.

**Statistical analysis.** Clinicopathologic data were abstracted from the electronic medical record. Overall response rate was calculated as the proportion of patients with either complete (CR) or partial (PR) response to ALK inhibitors. Progression-free survival was determined from the start date of ALK inhibition until the date of disease progression or death, and overall survival was calculated from the date of diagnosis of advanced NSCLC until the date of death. All p-values are two sided and confidence intervals are at the 95% level. Categorical variables were evaluated using Fisher's exact test or chi-square test, as appropriate. Continuous variables were evaluated with the Mann Whitney test, or Kruskal Wallis test, as appropriate. Event-time distributions were estimated using Kaplan-Meier methodology. Univariate and multivariate hazard ratios (HR) were assessed using the Cox proportional hazard model. All p-values are two sided, unless otherwise specified, and confidence intervals are at the 95% level, with significance pre-defined to be at $p < 0.05$.

**Statistical analysis**

The statistical analysis represented mean ± s.d. from three or more independent experiments. Data were analyzed by unpaired t-test for group differences and by two-way analysis of variance analysis of variance for condition and group differences together using GraphPad Prism v7.03 software. The statistical significance for comparing mRNA expression or peaks' scores of ATAC-seq and ChIP-seq was calculated by two-tailed Welch's t test in R3.6.

**Reporting summary**

Further information on research design is available in the Nature Portfolio Reporting Summary linked to this article.

# Data availability

All sequencing data generated in this study have been deposited in the Gene Expression Omnibus (GEO) database under accession number GSE167155. RNA-seq data used in this study are available in the GEO database under accession number GSM4635290 [https://www.ncbi.nlm.nih.gov/geo/query/acc.cgi?acc=GSE153183]. ATAC-seq data used in this study are available in the GEO database under accession number GSM1904729. H3K9ac ChIP-seq data used in this study are available in the GEO database under accession number GSM2534660. H3K9me3 ChIP-seq data used in this study are available in the GEO database under accession number GSM1912806. H3K4me3 ChIP-seq data used in this study are available in the Encyclopedia of DNA Elements (ENCODE) database under accession number ENCSR441JWF. H3K27ac ChIP-seq data used in this study are available in the ENCODE database under accession number ENCSR769FOC. Source data are provided with this paper.

# Code availability

Source code for genomic event analysis tool (GEAT)[77] developed in our laboratory to perform the analysis has been deposited in the Zenodo database [https://doi.org/10.5281/zenodo.6592772].

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

## Acknowledgements

We thank Dr. Nancy Chamberlin and Dr. Jake June-Koo Lee for the critical reading and editing of the manuscript, Dr. Fabio Iannelli for helpful discussions, and the members of the Laboratory for Molecular Pediatric Pathology (LaMPP) in the Pathology Department at Boston Children's Hospital for their assistance with the fusion sequencing. We thank Drs. Luisella Righi and Riccardo Taulli for providing data about one ALK + NSCLC case and for the helpful discussion. This work was supported by Ellison Foundation Boston and National Institutes of Health (NIH) 1R01-CA222598-01 to R.C.; the National Research Foundation of Korea (NRF) fellowship 2016R1A6A3A03006840 to T-C.C.; the OFD/BTREC/CTREC Faculty Career Development Fellowship to T-C.C.; NIH T32 DK007516 (A.J.); and the New York Stem Cell Foundation to M.K.L. M.K.L. is a New York Stem Cell Foundation—Robertson Investigator. Supplementary Fig. 1 is created with BioRender.com.

## Author contributions

T.-C.C. and A.J. performed most of the experiments and analysis. T.-C.C. and Q.W. performed bioinformatics analysis. B.R., J.V.A., and A.D.F analyzed the patient samples. T.-C.C. and G.C.L. performed mouse experiments. M.K.L. contributed with reagents and supervision. M.M.A. supervised the analysis of patient samples. M.H.H. performed analysis of 3' end-directed fusion assay. T.-C.C. and R.C. conceived the project, designed all experiments, analyzed, and interpreted the data, designed the figures, and wrote the manuscript. All authors edited the manuscript.

## Competing interests

The authors declare no competing interests.
