## [Peer Review File · Nature Communications]

Mechanistic patterns and clinical implications of oncogenic tyrosine kinase fusions in human cancersREVIEWER COMMENTS

Reviewer #1 (Remarks to the Author): expertise in tyrosine kinase fusions

Cheong et al address several mechanistic concepts relevant for the evolution and transforming activity of oncogenic kinase fusions. The work is experimentally solid and provides significant advances in our understanding of how fusion partners for kinases are selected and what molecular characteristics promote their enrichment during the evolution of functional kinase fusions, in particular in the context of lung cancer.

While the work is mostly based on in vitro analyses of two cell lines (the EGFR mutant PC-9 and the non-transformed BEAS-2B cell line), mouse xenograft assays as well as analysis of the COSMIC database and of 108 clinical lung cancer samples also provide in vivo and clinical relevance for the findings. The manuscript is well written to summarize the observations in a logical manner. While the number of clinical tissue samples representing the so called atypical ALK fusions is small (n = 11) limiting the power of statistical analyses, the conclusions drawn throughout the manuscript are mostly based on the data presented. The developed pipeline (FACTS) to screen for functional kinase fusions in vitro and in a xenograft model is novel.

This reviewer has only minor comments:

- 1) In Suppl. Table 2 the numbering of the tissues is off by 1 starting from the tissue number 4.
- 2) On p. 8, the description of the effects of lorlatinib vs osimertinib/lorlatinib combination on ERK phosphorylation and growth (referring to Figs 2c and d) could be improved to better explain single agent vs combinatory effects.
- 3) On p. 13, description of the HTGTS analysis before and after selection pressure (referring to Ext Data Fig. 6a) should include the information that ALK_{in}19 DSB was introduced at the start of the experiment.
- 4) On the p. 20 of Discussion, there is a sentence "Gene transcription was required not only to express the resulting fusion but also to increase the probability of DSBs occurring within a gene". The data presented, however, seem to support a hypothesis that up-regulated transcription associates with occurrence of DSBs (consistent with being "sufficient") but not formally the need (being "necessary" or "required") for the process.

Reviewer #3 (Remarks to the Author): expertise in fusion identification bioinformatics

Reviewer expertise: Cancer genomics, oncogenic fusions (no expertise on colony formation assay, IC50, and protein assay such as Western blots)

Re: "Mechanistic patterns and clinical implications of oncogenic tyrosine kinase fusions in human cancers" by Chiarle et al

In this nice work, the authors aim to study the underlying mechanism of oncogenic fusion formation, by utilizing a nice model system PC-9 that, upon EGFR inhibition (a strong selection pressure), is obligated to produce alternative oncogenic drivers such as ALK fusions. They then designed gRNAs to gene regions of interest to produce breaks, which will lead to significantly enhanced rate of translocation between the CRISPR target and other regions (or another CRISPR target if designed). They used NGS methods to measure the frequency of translocations and compare it between before- and after- selection. Under this nice experimental system, they reached following conclusions: typical TK fusions are selected from large pool of rearrangements, with two major determinants: active expression of the N terminus partner genes and protein stability. They validate their conclusion using patient data by showing that patients with atypical TK fusions are less responsive to TKI therapies.

I feel this paper is well written, I tentatively agree with their opinion on the mechanism of

oncogenic fusion formation is largely “random” (before selection) for TK fusions at the least, with a few additional molecular factors including abundance (i.e., the promoter activity of N terminus partner genes) and potentially protein stability. I like the neat idea of using cell line PC-9 that we can apply selection pressure on to enrich/enable clonal formation which is in general difficult to achieve.

I have following major comments:

1. There is a similar thinking in recent literature (PMID 37019972) where promoter activity, breakpoint randomness, intron/exon relevance, differential selection pressure (i.e., oncogenicity) were extensively investigated using patient data. The authors should compare their in vitro experiment data with patient-data based conclusions in the literature.
2. I could not find a detailed description of FACTS in method section. I understand this is a combination of gRNA-based DSB generation; EGFR inhibition-based selection; target gene as PCR anchor; NGS and analysis. But it might be helpful to summarize these to a method section or a figure?
3. Line #351, would it be possible to see if the clonal formation is also increased when MG132 is used? This is in conjunction with my comment on line #365 on E6;A18.
4. Line #365, the existence of patients with E6;A18 kind of argues against the “lack of protein stability” result in line #351-#352. Please elaborate. Basically, based on these data, this reviewer believes that there might be additional molecular features rather than stability to explain the “lower” oncogenicity of E6;A18.
5. Line #386, if only E6;A20 fusion is stable enough, then there should NOT be patients with E6;A18 (line #365). Could the data of E6;A18 being wrong instead (which is against the proposal that there is additional molecular factors but can equally explain the observation)?
6. Line #449-450. It is stated that this method is good when a method of selection pressure is available. How could it be generalized to non-TK fusions and other translocations? I worry this speculation is not backed by data.
7. Line #458. It was mentioned that the DSB is even in line #318. It is not very inline with this bias toward high expression genes.
8. I am not sure if I can see the pattern in Fig. 3c (therefore no scientific judgement). Some kind of density plot on top of this panel might be helpful? Because this data is centered around ALK, I think the authors should group the color by whether it would be a biologically meaningful fusion with ALK, rather than just by strand? Fine to keep using the colors as-is if the authors decide so.

Minor comments:

1. Line #135, please add citation to the “recent reports”. Are they really “recent”?
2. Line #145, the citation should add #7---the material foundation of this work.
3. Fig. 3b, please define “Alignment Score” in Method section. I could not make any scientific judgement on this important figure due to lack of details.
4. Line #177, it would be great to provide (rough) estimated number of total cells during the process so that readers can have a feeling of the chance of the host cells to spontaneously produce the “correct” DNA breakpoints. These numbers should be separated for single guide and double guide scenarios.
5. Line #318, please provide a quantification of “evenness” if possible. Consider using the statistical test of Fig. 2h in PMID 37019972 (or whatever the authors consider good).

Reviewer #4 (Remarks to the Author): expert in tyrosine kinase in vivo biology

Summary

This paper provides intriguing insights into the biology that underlies the development of kinase fusion oncogenes in human cancers. Focusing on ALK, RET, ROS1, NTRK, ABL1 they establish an experimental system to interrogate the development and functional roles of fusion oncogenes arising from these kinases. The screening assay uses crispr/cas9 to induce DSB at select genomic

locations and the assay then selects for the development of EGFR-resistance in PC9 cells or the development of tumorigenesis in BEAS-2B bronchial epithelial cells and thus selects for functional disease drivers that have been promoted by genome targeting. The experimental system does seem to replicate many of the fusion events seen in human cancers and reports many interesting findings. The exon selection is not random and there is significant preference for fusion at certain exons in certain TKs, and this has to do with protein stability issues. The expression of the 5' fusion partner is important and this is confirmed in experiments with silent partners that can be experimentally induced to express. The partner-TK pairings are also not random, and there are preferential pairings. A particularly important translational implication is regarding atypical fusions. Specifically with regards to ALK, they show that in addition to the common ex20 fusions with potent oncogene formation, there are less common atypical fusions at other exons, and these have less signaling potency, and they are not the dominant disease drivers that the ex20 fusions are. Tumorigenicity in these cases is likely due to combined functions of the ALK fusion and other driver events, and accounts for the reduced activity of ALK inhibitor monotherapy in these tumors.

General comments

This is a solid piece of experimental science used to explain many of the observations reported in human cancers regarding TK fusions. There is a lot of work done here, the experiments are well designed, and the data is convincing. The manuscript is well written, the figures are well prepared, and the discussion is relevant and informative. I don't have any major criticisms to offer, and would support the publication of this work. I provide some minor suggestions below as constructive criticism, should they find it helpful.

Minor criticisms

Most of the data is in the extended figures. I believe they are well under the figure limits for Nat Comm. This distribution of data/figures may be a carryover effect from prior submissions to other journals. But I think much more of the data should be in the main body of figures.

Page 10 describes the use of FACTS to induce ROS1 fusions. They induce DSB intron 33 of ROS1, but the fusion partners identified involve ROS1 fusions at exons 34,35, and 36. Does the FACTS technique allow wobble around the cutting site? Is it not precise? Please provide an explanation regarding the FACTS technique and the observed findings.

The atypical ALK fusion cases are less responsive to TKI therapy as shown in figure 6a,b. The experimental work in this study shows that these have lower expression and lower signaling potency. Thus it is likely that they are not as dominant a disease driver as are the typical fusions. Figure 6c provides tumor genetic data with the contention that the atypical cases have additional disease drivers. This hypothesis is likely true and the data may be consistent with it, but the number of atypical fusion cases is really too small for firm conclusions. I would just tone down the verbiage given that the data set is very small. Afterall, the experimental work with the atypical fusions is also limited. They did not test these atypical fusions in BEAS-2B cell model with transformation assays or in vivo tumorigenicity studies to show that they do have any tumorigenic activities. An alternative explanation to the "co-operating drivers: hypothesis is that some of these atypical fusions may be even weaker than they think. They may be "passenger" fusion events, basically irrelevant, similar to passenger mutations seen in many genes.

RESPONSE TO REVIEWERS' COMMENTS

Reviewer #1

Cheong et al address several mechanistic concepts relevant for the evolution and transforming activity of oncogenic kinase fusions. The work is experimentally solid and provides significant advances in our understanding of how fusion partners for kinases are selected and what molecular characteristics promote their enrichment during the evolution of functional kinase fusions, in particular in the context of lung cancer.

While the work is mostly based on in vitro analyses of two cell lines (the EGFR mutant PC-9 and the non-transformed BEAS-2B cell line), mouse xenograft assays as well as analysis of the COSMIC database and of 108 clinical lung cancer samples also provide in vivo and clinical relevance for the findings. The manuscript is well written to summarize the observations in a logical manner. While the number of clinical tissue samples representing the so called atypical ALK fusions is small (n = 11) limiting the power of statistical analyses, the conclusions drawn throughout the manuscript are mostly based on the data presented. The developed pipeline (FACTS) to screen for functional kinase fusions in vitro and in a xenograft model is novel.

[Response] We appreciate this Reviewer's overall positive comments.

This reviewer has only minor comments:

1) In Suppl. Table 2 the numbering of the tissues is off by 1 starting from the tissue number 4.

[Response] Thank you for finding this mistake. We have corrected the Table in the revised version of the manuscript: "1-Biliary tract; 2-Breast; 3-Central nervous system; 4-Haematopoietic and lymphoid tissue; 5-Kidney; 6-Large intestine; 7-Lung; 8-Ovary; 9-Pancreas; 10-Peritoneum; 11-Prostate; 12-Salivary gland; 13-Skin; 14-Soft tissue; 15-Stomach; 16-Thyroid".

2) On p. 8, the description of the effects of lorlatinib vs osimertinib/lorlatinib combination on ERK phosphorylation and growth (referring to Figs 2c and d) could be improved to better explain single agent vs combinatory effects.

[Response] We have revised the manuscript to improve clarity with the following edited sentences "The ALK fusions formed after the DSBs at *ALK* intron 19 showed strong phosphorylation of the kinase domain, resulting in sustained activation of the MAPK pathway despite the presence of osimertinib, but showed impaired phosphorylation of the kinase domain in the presence of ALK-specific inhibitor lorlatinib, resulting in decreased activation of the MAPK pathway (**Fig. 2c**), likely explaining the mechanism of resistance. The combination of osimertinib with lorlatinib completely blocked ALK and ERK1/2 phosphorylation (**Fig. 2c**). Consistently, the growth of osimertinib-resistant clones was inhibited by the combination of osimertinib with lorlatinib, but not by osimertinib or lorlatinib alone (**Fig. 2d**)".

3) On p. 13, description of the HTGTS analysis before and after selection pressure (referring to Ext Data Fig. 6a) should include the information that ALK_{in19} DSB was introduced at the start of the experiment.

[Response] We added this information in the revised manuscript as following: "To this end, we generated libraries of DNA junctions using HTGTS, allowing for an unbiased detection of genome-wide chromosomal rearrangements¹, in PC-9 cells introduced a programmed DSB in intron 19 of *ALK* in both

before and after selection (**Supplementary Fig. 6a, b**)”.

4) On the p. 20 of Discussion, there is a sentence "Gene transcription was required not only to express the resulting fusion but also to increase the probability of DSBs occurring within a gene". The data presented, however, seem to support a hypothesis that up-regulated transcription associates with occurrence of DSBs (consistent with being "sufficient") but not formally the need (being "necessary" or "required") for the process.

[Response] We agree with the Reviewer on this interpretation and corrected the manuscript: “Gene transcription was sufficient not only to express the resulting fusion but also to increase the probability of DSBs occurring within introns and exons of a gene with increased transcription”.

Reviewer #3

In this nice work, the authors aim to study the underlying mechanism of oncogenic fusion formation, by utilizing a nice model system PC-9 that, upon EGFR inhibition (a strong selection pressure), is obligated to produce alternative oncogenic drivers such as ALK fusions. They then designed gRNAs to gene regions of interest to produce breaks, which will lead to significantly enhanced rate of translocation between the CRISPR target and other regions (or another CRISPR target if designed). They used NGS methods to measure the frequency of translocations and compare it between before- and after- selection. Under this nice experimental system, they reached following conclusions: typical TK fusions are selected from large pool of rearrangements, with two major determinants: active expression of the N terminus partner genes and protein stability. They validate their conclusion using patient data by showing that patients with atypical TK fusions are less responsive to TKI therapies.

I feel this paper is well written, I tentatively agree with their opinion on the mechanism of oncogenic fusion formation is largely “random” (before selection) for TK fusions at the least, with a few additional molecular factors including abundance (i.e., the promoter activity of N terminus partner genes) and potentially protein stability. I like the neat idea of using cell line PC-9 that we can apply selection pressure on to enrich/enable clonal formation which is in general difficult to achieve.

[Response] Thank you very much for taking the time and effort to thoroughly review our manuscript. We appreciate the constructive comments and suggestions.

I have following major comments:

1. There is a similar thinking in recent literature (PMID 37019972) where promoter activity, breakpoint randomness, intron/exon relevance, differential selection pressure (i.e., oncogenicity) were extensively investigated using patient data. The authors should compare their in vitro experiment data with patient-data based conclusions in the literature.

[Response] We appreciate this Reviewer’s suggestion. The indicated study (PMID: 37019972) comprehensively analyzed sequencing data from 5,190 childhood cancers, including leukemia, brain tumor, and solid tumor, and identified 2,012 oncogenic fusion events (272 gene pairs from 2,005 fusion-positive patients). By analyzing a large datasets of tumor transcriptome sequencing data, they categorized the fusion events as neo-translational, intron-versioning, neo-splicing, chimeric exon and found several factors that influence the formation of oncogenic gene fusions, such as translation frame, protein domain, splicing, and gene length. We applied these criteria to our FACTS datasets and identified events

consistent with intronic versioning (i.e., EML4-ALK, TFG-ALK) and neo-splicing (i.e., TPM3-NTRK1), but not neo-translational or chimeric exon events because we targeted specific introns of 3' tyrosine kinase (TK) genes with CRISPR-Cas9 guides. We could not see a significant correlation between gene length and oncogenic *ALK* fusions (**Figure for Reviewer only**).

Figure for Reviewer only. No correlation showed between gene length of *ALK* fusion partner genes ($n = 12$) and breakpoint frequency in PC-9 cells after selection.

Consistent with the findings in the literature, our FACTS data showed that the breakpoints of fusion partners are enriched for specific introns leading to fusions containing specific exons that are positively selected as oncogenic fusions in patients (i.e., EML4 introns 2, 6, 13, and 18; **Fig. 2e**). The advantage of our experimental approach compared to studies that analyze patient samples is that we experimentally demonstrated that *EML4* fusions with different *ALK* exons 18, 19, and 20 are all predicted to be in-frame and can be formed, but display different oncogenicity (**Fig. 7**). This different oncogenicity provides an explanation for the overwhelming prevalence of fusion involving *ALK* exon 20 in patients (**Fig. 7**), which could not be anticipated by previous works on samples from patients.

To summarize these findings and comparisons, we included the following statement in the Discussion of the revised manuscript: “Comprehensive analysis of transcriptome sequencing data on gene fusions in childhood cancers⁵ has shown multiple molecular factors that influence oncogenic gene fusions, including, gene length, splicing translation frame, and protein domains. Our experimental model provided evidence for some of these mechanisms, such as intron-versioning and neo-splicing⁵, while also providing evidence for a selection process that enriches for specific TK fusions in human cancers.”

2. I could not find a detailed description of FACTS in method section. I understand this is a combination of gRNA-based DSB generation; EGFR inhibition-based selection; target gene as PCR anchor; NGS and analysis. But it might be helpful to summarize these to a method section or a figure?

[Response] We agree that a better description of the FACTS approach is warranted. To this end, we added a schematic overview of the FACTS approach in the revised version to rapidly capture the basis of FACTS (**Supplementary Fig. 1**). FACTS was developed to selectively identify functional fusions in *in vitro* and *in vivo* experimental conditions leveraging the enrichment of oncogenic events by selective pressure. We employed an EGFR-mutant non-small cell lung cancer cell line PC-9, which is dependent on EGFR signaling and sensitive to an EGFR inhibitor (as an *in vitro* model), and a non-tumorigenic bronchial lung epithelial cell line BEAS-2B, which is only grown in mice when transformed with an oncogenic driver (as an *in vivo* model). The cells were first introduced DNA DSBs in a specific genomic location (e.g., tyrosine kinase gene) by the CRISPR/Cas9 system and were selected with an EGFR inhibitor (for the *in vitro* PC-9 cell model) and in mice (for the *in vivo* BEAS-2B cell model). The resistant clones to the EGFR inhibitor in PC-9 cells (*in vitro*) and mouse tumors (*in vivo*) were collected, and total nucleic acid (TNA) was extracted. The library was prepared with using the Archer FusionPlex library kit with custom gene-specific primers that include *ALK*, *RET*, *ROS1*, and *NTRK1*. The libraries were sequenced using an Illumina MiSeq sequencer. Sequences were analyzed using ArcherAnalysis

bioinformatics software and manually confirmed for frames and domain structures. This information is now added for a better description of FACTS in the revised method section.

Supplementary Fig. 1. Overview of FACTS approach. Cells driven by specific oncogenic driver will be chosen, generated DNA DSBs in a gene of interest, and selected with inhibitors of the specific oncogenic driver. Library will be prepared using gene-specific primers and sequenced using illumina MiSeq. Sequence data will be analyzed with Archer and manual analysis (see Methods). TKI, tyrosine kinase inhibitor.

3. Line #351, would it be possible to see if the clonal formation is also increased when MG132 is used? This is in conjunction with my comment on line #365 on E6;A18.

[Response] We thank the Reviewer for this suggestion. Indeed, this experiment came to our mind, and it was already attempted. The problem we faced is that MG132 inhibits cell growth, causes cell cycle arrest at the G2 phase, and promotes apoptosis when the treatment is administered for prolonged time⁶⁻⁹. For FACTS, we need resistant clones to proliferate and expand for several weeks to be detected and enriched. As a matter of fact, the cells (E6;A18, E6;A19, and E6;A20) were treated with MG132 for just 18 hours, yet they all showed significant growth inhibition compared to untreated control.

4. Line #365, the existence of patients with E6;A18 kind of argues against the “lack of protein stability” result in line #351-#352. Please elaborate. Basically, based on these data, this reviewer believes that there might be additional molecular features rather than stability to explain the “lower” oncogenicity of E6;A18.

Patients with E6;A18 or E6;A19 do exist but are very rare compared to EML4-ALK fusion involving exon 20 that represent >95% of events. In this work, we demonstrate that this preference for ALK exon 20 cannot be explained by DSB distribution, by mRNA expression, abundance, splicing or stability, or by protein localization, given the importance membraneless cytoplasmic receptor tyrosine kinase (RTK) fusion granule formation for a robust oncogenic signaling in cells¹⁰ (**Supplementary Fig. 8**). The only marked difference we observed was a lower protein abundance associated with weaker downstream

oncogenic signaling, likely due to diminished stability. Nonetheless, we cannot exclude that other reasons could contribute to the lower oncogenicity of E6;A18 or E6;A19 fusions.

In the revised Discussion we commented on the possibility that other factor could explain the selection of specific TK fusion: “Thus, the selection process of the specific fusion partner for each TK depends not only on the availability of an in-frame dimerization domain but also on the stability of the resulting fusion. Furthermore, the pattern of recurrent oncogenic TK fusions may be dictated by the protein stability of the resulting fusion proteins rather than the enriched DSBs in specific locations in the cancer genome. Yet, we cannot exclude that other mechanisms other than protein stability could contribute to the selection of the oncogenic TK fusions in cancers.”

5. Line #386, if only E6;A20 fusion is stable enough, then there should NOT be patients with E6;A18 (line #365). Could the data of E6;A18 being wrong instead (which is against the proposal that there is additional molecular factors but can equally explain the observation)?

[Response] As we shown with the COSMIC dataset (**Fig. 1e**), *ALK* exon 20 fusions are present more than 99% of cancer patients. In contrast, *ALK* exon18 and exon19 fusions are rare but still can be discovered in patients using DNA- and RNA-based sequencing techniques. In this manuscript, we investigated datasets of patient with NSCLC from Dana Farber Cancer Institute using OncoPanel (a DNA-based next-generation sequencing approach) to find *ALK* fusions. Our patient data revealed multiple occurrences of E6;A18 and E6;A19 fusions (**Supplementary Fig. 9**), but their frequency was very low as expected from literature data (less than 4%, 11 out of 300 cases). This data are in keeping with the RNA-based sequencing data reported in the COSMIC dataset where E6;A18 and E6;A19 fusions are reported to be less than 1% of all EML4-*ALK* fusions. (**Fig. 1e**). Thus E6;A18 and E6;A19 fusions do exist in human cancer samples but are very rare. The published literature, mentioned in the manuscript, also confirms this strong bias.

6. Line #449-450. It is stated that this method is good when a method of selection pressure is available. How could it be generalized to non-TK fusions and other translocations? I worry this speculation is not backed by data.

[Response] By this statement, we argue that FACTS could be used not only to enrich for TK fusion that drive resistance under selective pressure, but also for other oncogenic event (different from TK fusions) that might have the same oncogenic potential to induce clone expansion under selective pressure. We agree that currently we do not have data to back this speculation. Therefore, we edited this sentence in the revised manuscript to better elaborate on this point and be clear that this is a possibility to be demonstrated by experimental evidence as following “In principle, the FACTS approach could be possibly expanded to study the formation and selection of other chromosomal rearrangements different from TK fusions occurring in sarcomas, hematologic malignancies, or other tumors, but it needs to be demonstrated by experimental evidence.”

7. Line #458. It was mentioned that the DSB is even in line #318. It is not very inline with this bias toward high expression genes.

[Response] In line #318, we claim that DSBs are evenly distributed within individual genes in introns and exons. In line #458, we claim that changes in transcription of a gene can modify the frequency of DSBs occurring in that gene. The two concepts are different, but we understand that there could be some confusion. To clarify this point we revised the manuscript to edit line #458: “Gene transcription was sufficient not only to express the resulting fusion but also to increase the probability of DSBs occurring within introns and exons of a gene with increased transcription”.

8. I am not sure if I can see the pattern in Fig. 3c (therefore no scientific judgement). Some kind of density plot on top of this panel might be helpful? Because this data is centered around ALK, I think the authors should group the color by whether it would be a biologically meaningful fusion with ALK, rather than just by strand? Fine to keep using the colors as-is if the authors decide so.

[Response] Thank you for your comment that helped improving the explanation for this figure. We revised the **Figure 5** legend as following “**b and c**, Rainfall plots showing genome-wide distribution of translocations before (**b**) and after (**c**) osimertinib selection in PC-9 cells. Each dot represents a single DNA translocation ordered on the X-axis according to its position in the human genome. Red and blue dots represent the orientation of DNA translocations on chromosome plus or minus strand, respectively. Example genes that transcribe in the plus and minus directions are shown in blue and red, respectively, resulting in functional *ALK* fusions in the correct direction. Data pooled from six biological replicates”.

Minor comments:

1. Line #135, please add citation to the “recent reports”. Are they really “recent”?

[Response] Thank you for pointing this out. We included three recent publications¹¹⁻¹³: Offin M et al., *JCO Precis Oncol.* 2018. PMID: 30957057; Kobayashi Y et al., *Nat Commun.* 2022. PMID: 36153311; Schrock AB et al., *J Thorac Oncol.* 2018. PMID: 29883838.

2. Line #145, the citation should add #7---the material foundation of this work.

[Response] We added the citation¹³ (Offin M et al., *JCO Precis Oncol.* 2018. PMID: 30957057).

3. Fig. 3b, please define “Alignment Score” in Method section. I could not make any scientific judgement on this important figure due to lack of details.

[Response] In the revised Method section we now clarify that the “Alignment Score” is BLAT SCORE generated by BLAT tool and calculated by the number of matches with a penalty for mismatches and gaps.

4. Line #177, it would be great to provide (rough) estimated number of total cells during the process so that readers can have a feeling of the chance of the host cells to spontaneously produce the “correct” DNA breakpoints. These numbers should be separated for single guide and double guide scenarios.

[Response] This is a good point. We provided the frequency of resistant clone development in **Supplementary Fig. 2d** (double DSB scenarios; estimated frequencies were around 1×10^4 clones/million cells by targeting both *EML4* and *ALK*) and **Fig. 2b** (single DSB scenarios; estimated frequencies were around 0.05 or 1 clone/million cells by targeting only *EML4* intron 6 or *ALK* intron 19, respectively).

5. Line #318, please provide a quantification of “evenness” if possible. Consider using the statistical test of Fig. 2h in PMID 37019972 (or whatever the authors consider good).

[Response] We understand this point and agree it is difficult to determine an “evenness score”. Rather we can exclude significant clusters of increased DSB frequency. Thus, we revised the statement as following “Breakpoints identified in these kinases were spread throughout the gene body including introns and exons without clear clusters (**Supplementary Fig. 7b-j**)”. We removed the word “evenly” from the sentence.

Reviewer #4

Summary

This paper provides intriguing insights into the biology that underlies the development of kinase fusion oncogenes in human cancers. Focusing on ALK, RET, ROS1, NTRK, ABL1 they establish an experimental system to interrogate the development and functional roles of fusion oncogenes arising from these kinases. The screening assay uses crispr/cas9 to induce DSB at select genomic locations and the assay then selects for the development of EGFR-resistance in PC9 cells or the development of tumorigenesis in BEAS-2B bronchial epithelial cells and thus selects for functional disease drivers that have been promoted by genome targeting. The experimental system does seem to replicate many of the fusion events seen in human cancers and reports many interesting findings. The exon selection is not random and there is significant preference for fusion at certain exons in certain TKs, and this has to do with protein stability issues. The expression of the 5' fusion partner is important and this is confirmed in experiments with silent partners that can be experimentally induced to express. The partner-TK pairings are also not random, and there are preferential pairings. A particularly important translational implication is regarding atypical fusions. Specifically with regards to ALK, they show that in addition to the common ex20 fusions with potent oncogene formation, there are less common atypical fusions at other exons, and these have less signaling potency, and they are not the dominant disease drivers that the ex20 fusions are. Tumorigenicity in these cases is likely due to combined functions of the ALK fusion and other driver events, and accounts for the reduced activity of ALK inhibitor monotherapy in these tumors.

General comments

This is a solid piece of experimental science used to explain many of the observations reported in human cancers regarding TK fusions. There is a lot of work done here, the experiments are well designed, and the data is convincing. The manuscript is well written, the figures are well prepared, and the discussion is relevant and informative. I don't have any major criticisms to offer, and would support the publication of this work. I provide some minor suggestions below as constructive criticism, should they find it helpful.

[Response] We thank this Reviewer for the overall summary of the key findings and all the positive general comments.

Minor criticisms

Most of the data is in the extended figures. I believe they are well under the figure limits for Nat Comm. This distribution of data/figures may be a carryover effect from prior submissions to other journals. But I think much more of the data should be in the main body of figures.

[Response] We appreciate this comment that will improve the outline of the manuscript with important data shown in main Figures. In the revised manuscript, we rearranged the Figures into 8 Main Figures and 10 Supplementary Figures.

Page 10 describes the use of FACTS to induce ROS1 fusions. They induce DSB intron 33 of ROS1, but the fusion partners identified involve ROS1 fusions at exons 34,35, and 36. Does the FACTS technique allow wobble around the cutting site? Is it not precise? Please provide an explanation regarding the FACTS technique and the observed findings.

[Response] We agree with this comment that gives the opportunity to better explain the FACTS approach. We added a schematic overview of the FACTS approach in the revised version (**Supplementary Fig. 1**). The FACTS approach was developed to selectively identify functional fusions in *in vitro* and *in vivo* settings in which selective pressure is available. We employed an EGFR-mutant non-small cell lung cancer cell line PC-9, which is dependent on EGFR signaling and sensitive to an EGFR inhibitor (as an *in vitro* model), and a non-tumorigenic bronchial lung epithelial cell line BEAS-2B, which is only grown in mice when transformed with an oncogenic driver (as an *in vivo* model). The cells were first introduced DNA DSBs in a specific genomic location (e.g., tyrosine kinase gene) by the CRISPR/Cas9 system and were selected with an EGFR inhibitor (for the *in vitro* PC-9 cell model) and in mice (for the *in vivo* BEAS-2B cell model). The resistant clones to the EGFR inhibitor in PC-9 cells (*in vitro*) and mouse tumors (*in vivo*) were collected, and total nucleic acid (TNA) was extracted. The library was prepared with using the Archer FusionPlex library kit with custom gene-specific primers that include *ALK*, *RET*, *ROSI*, and *NTRK1*. The libraries were sequenced using an Illumina MiSeq sequencer. Sequences were analyzed using ArcherAnalysis bioinformatics software and manually confirmed for frames and domain structures.

Supplementary Fig. 1. Overview of FACTS approach. Cells driven by specific oncogenic driver will be chosen, generated DNA DSBs in a gene of interest, and selected with inhibitors of the specific oncogenic driver. Library will be prepared using gene-specific primers and sequenced using illumina MiSeq. Sequence data will be analyzed with Archer and manual analysis (see Methods). TKI, tyrosine kinase inhibitor.

The primers for fusion detection are designed to sequence several exons frequently involved in kinase gene fusions. In the case of *ROSI*, the Archer FusionPlex library includes primers that detect exons 34, 35, and 36. Thus, while we generated DSB in intron 33 of *ROSI*, FACTS captured *ROSI* fusions involving not only the expected exon 34, but also fusion involving exons 35 and 36. This shift of exons is likely occurring during the process of DNA repair and translocation formation that is known to extend and spread for several kilobases flanking the original DSB due to resection of the DNA broken ends^{14,15}. Of note, this spreading was observed with FACTS with *ROSI* and *NTRK1*, but not with *ALK* and *RET*. This finding is consistent with data in patients that indicated that *ROSI* and *NTRK1* oncogenic fusion can involve multiple *ROSI* and *NTRK1* exons (**Fig. 1e**), while *ALK* and *RET* fusion are much more restricted

to specific exons (exon 20 for *ALK* and exon 12 for *RET*). Thus, we interpreted this exon spreading for certain TK is not a lack of precision of the FACTS approach but rather an added value due to the potential to capture oncogenic fusions that spread in the flanking exons during the DNA repair process. In the revised manuscript, we further clarified this point as following “Since translocation formation is known to extend and spread for several kilobases flanking the original DSB due to recession of the DNA broken ends during the DNA repair process^{14,15}, FACTS allows to detect exon fusion variants of TK fusions by designing sequencing primers in multiple exons for fusion detection.”

The atypical *ALK* fusion cases are less responsive to TKI therapy as shown in figure 6a,b. The experimental work in this study shows that these have lower expression and lower signaling potency. Thus it is likely that they are not as dominant a disease driver as are the typical fusions. Figure 6c provides tumor genetic data with the contention that the atypical cases have additional disease drivers. This hypothesis is likely true and the data may be consistent with it, but the number of atypical fusion cases is really too small for firm conclusions. I would just tone down the verbiage given that the data set is very small. After all, the experimental work with the atypical fusions is also limited. They did not test these atypical fusions in BEAS-2B cell model with transformation assays or in vivo tumorigenicity studies to show that they do have any tumorigenic activities. An alternative explanation to the “co-operating drivers: hypothesis is that some of these atypical fusions may be even weaker than they think. They may be “passenger” fusion events, basically irrelevant, similar to passenger mutations seen in many genes.

[Response] We agree that the number of atypical fusions is too low to allow for definitive conclusions. Yet, these atypical fusions are quite rare, and we needed to start with a very large cohort of NSCLC to get to this number. Our point is that these atypical fusions may be very weak drivers, but not just passenger events because several patients with these atypical fusions showed partial and transient response to *ALK* targeted therapy (**Supplementary Table 8**). It is known that patients with *ALK*-negative NSCLC are unresponsive to *ALK* TKI. Therefore, if these atypical *ALK* fusion would be only passenger, we would expect a lack of any clinical response in these patients to *ALK* TKI. We revised the discussion to comment on this important point as following “Because patients with atypical *EML4-ALK* fusions showed partial and transient response to *ALK* targeted therapy (**Supplementary Fig. 9d and Supplementary Table 8**), these atypical fusions may be very weak oncogenic drivers”.

References

1. Chiarle, R., *et al.* Genome-wide translocation sequencing reveals mechanisms of chromosome breaks and rearrangements in B cells. *Cell* **147**, 107-119 (2011).
2. Stransky, N., Cerami, E., Schalm, S., Kim, J.L. & Lengauer, C. The landscape of kinase fusions in cancer. *Nat Commun* **5**, 4846 (2014).
3. Gao, Q., *et al.* Driver Fusions and Their Implications in the Development and Treatment of Human Cancers. *Cell Rep* **23**, 227-238 e223 (2018).
4. Vellichirammal, N.N., *et al.* Pan-Cancer Analysis Reveals the Diverse Landscape of Novel Sense and Antisense Fusion Transcripts. *Mol Ther Nucleic Acids* **19**, 1379-1398 (2020).
5. Liu, Y., *et al.* Etiology of oncogenic fusions in 5,190 childhood cancers and its clinical and therapeutic implication. *Nat Commun* **14**, 1739 (2023).
6. Guo, N. & Peng, Z. MG132, a proteasome inhibitor, induces apoptosis in tumor cells. *Asia Pac J Clin Oncol* **9**, 6-11 (2013).

7. MacLaren, A.P., Chapman, R.S., Wyllie, A.H. & Watson, C.J. p53-dependent apoptosis induced by proteasome inhibition in mammary epithelial cells. *Cell Death Differ* **8**, 210-218 (2001).
8. Wente, M.N., *et al.* The proteasome inhibitor MG132 induces apoptosis in human pancreatic cancer cells. *Oncol Rep* **14**, 1635-1638 (2005).
9. Westerberg, C.M., Hagglund, H. & Nilsson, G. Proteasome inhibition upregulates Bim and induces caspase-3-dependent apoptosis in human mast cells expressing the Kit D816V mutation. *Cell Death Dis* **3**, e417 (2012).
10. Tulpule, A., *et al.* Kinase-mediated RAS signaling via membraneless cytoplasmic protein granules. *Cell* **184**, 2649-2664 e2618 (2021).
11. Kobayashi, Y., *et al.* Genomic and biological study of fusion genes as resistance mechanisms to EGFR inhibitors. *Nat Commun* **13**, 5614 (2022).
12. Schrock, A.B., *et al.* Receptor Tyrosine Kinase Fusions and BRAF Kinase Fusions are Rare but Actionable Resistance Mechanisms to EGFR Tyrosine Kinase Inhibitors. *J Thorac Oncol* **13**, 1312-1323 (2018).
13. Offin, M., *et al.* Acquired ALK and RET Gene Fusions as Mechanisms of Resistance to Osimertinib in EGFR-Mutant Lung Cancers. *JCO Precis Oncol* **2**(2018).
14. Lee, J.J., *et al.* ERalpha-associated translocations underlie oncogene amplifications in breast cancer. *Nature* **618**, 1024-1032 (2023).
15. Compagno, M., *et al.* Phosphatidylinositol 3-kinase delta blockade increases genomic instability in B cells. *Nature* **542**, 489-493 (2017).

REVIEWERS' COMMENTS

Reviewer #1 (Remarks to the Author):

The authors have appropriately addressed all the comments by this reviewer.

Reviewer #3 (Remarks to the Author):

The authors have thoroughly addressed my concerns and congratulations on this nice work!

I really like the innovative experimental approach to investigate molecular mechanisms that shape the formation of oncogenic fusions by teasing out selection pressure, and the discovery of novel molecular mechanism of protein stability, in addition to the differential oncogenicity that are also found from primary patient cohorts.

Reviewer #4 (Remarks to the Author):

I only had suggestions in my original critiques, and the authors have appropriately responded and acted on them. I have no further critiques to offer.